# FAF1 phosphorylation by AKT accumulates TGF-β type II receptor and drives breast cancer metastasis

Feng Xie[1,*], Ke Jin[1,*], Li Shao[2,*], Yao Fan[1], Yifei Tu[1], Yihao Li[3], Bin Yang[4], Hans van Dam[3], Peter ten Dijke[3], Honglei Weng[5], Steven Dooley[5], Shuai Wang[6], Junling Jia[1], Jin Jin[1], Fangfang Zhou[6] & Long Zhang[1]

TGF-β is pro-metastatic for the late-stage breast cancer cells. Despite recent progress, the regulation of TGF-β type II receptor remains uncertain. Here we report that FAF1 destabilizes TβRII on the cell surface by recruiting the VCP/E3 ligase complex, thereby limiting excessive TGF-β response. Importantly, activated AKT directly phosphorylates FAF1 at Ser 582, which disrupts the FAF1–VCP complex and reduces FAF1 at the plasma membrane. The latter results in an increase in TβRII at the cell surface that promotes both TGF-β-induced SMAD and non-SMAD signalling. We uncover a metastasis suppressing role for FAF1 through analyses of FAF1-knockout animals, various *in vitro* and *in vivo* models of epithelial-to-mesenchymal transition and metastasis, an MMTV-PyMT transgenic mouse model of mammary tumour progression and clinical breast cancer samples. These findings describe a previously uncharacterized mechanism by which TβRII is tightly controlled. Together, we reveal how SMAD and AKT pathways interact to confer pro-oncogenic responses to TGF-β.

[1] Life Sciences Institute and Innovation Center for Cell Signalling Network, Hangzhou, Zhejiang 310058, China. [2] State Key Laboratory for Diagnostic and Treatment of Infectious Diseases, The First Affiliated Hospital, School of Medicine, Zhejiang University, Collaborative Innovation Center for Diagnosis and Treatment of Infectious Disease, Hangzhou 310000, China. [3] Department of Molecular Cell Biology, Cancer Genomics Centre Netherlands, Leiden University Medical Center, Postbus 9600 2300 RC Leiden, The Netherlands. [4] Department of Pharmaceutical Chemistry and the Cardiovascular Research Institute, University of California San Francisco, San Francisco, California 94158, USA. [5] Department of Medicine II, Medical Faculty Mannheim, Heidelberg University, Mannheim 105760, Germany. [6] Institutes of Biology and Medical Science, Soochow University, Suzhou 215123, China. * These authors contributed equally to this work. Correspondence and requests for materials should be addressed to L.Z. (email: L_Zhang@zju.edu.cn) or to F.Z. (email: zhoufangfang@suda.edu.cn).

Transforming growth factor-β (TGF-β) is a pro-metastatic factor in advanced cancer[1–4]. Upon ligand binding, the TGF-β type II serine/threonine kinase receptor (TβRII) activates the type I receptor (TβRI) to induce SMAD2/3 phosphorylation. Activated SMAD2/3 forms hetero-oligomers with SMAD4, which accumulate in the nucleus to regulate target genes[1–3]. In addition to the canonical SMAD pathway, TGF-β receptors can initiate other intracellular pathways via either phosphorylation or direct interaction with signalling intermediates; these so-called non-SMAD signalling pathways include several branches that involve phosphatidylinositol kinase (PI3K)/AKT, mitogen-activated protein kinases (MAPKs) and Rho-like GTPase signalling intermediates[5].

TGF-β cross-talks with other pathways[6]. Oncogenic PI3K/AKT signalling antagonizes TGF-β-induced growth arrest and apoptotic responses[7,8]. Moreover, high TGF-β levels in tumours correlate with overactive PI(3)K–AKT signalling, and poor prognosis in breast cancer[9–11]. However, how AKT cross-reacts with TGF-β-induced pro-invasive and pro-metastatic responses in advanced tumours remains undefined. In the TGF-β/SMAD canonical pathway, TβRI acts downstream of TβRII; the stability and membrane localization of TβRII are therefore critical determinants of both the sensitivity and duration of the TGF-β response. Many previous studies have concluded that TβRII mediates the cytostatic effects of TGF-β; loss of its function in many different cancer models promotes aggressive and metastatic behaviour[12,13]. Whether a gain of function in TβRII can promote metastasis has not been thoroughly investigated. In this work, we identify FAS-associated factor 1 (FAF1) as a key regulator of cell surface TβRII, in turn preventing the excessive activation of both SMAD and non-SMAD TGF-β-induced signalling. During cancer progression, growth factor-induced (or oncogenic mutation) activation of AKT mediates FAF1 phosphorylation and its dissociation from the plasma membrane and TβRII, thereby reinforcing TβRII stability on the cell surface and activating the pro-metastatic functions induced by TGF-β in breast cancer cells.

## Results

**FAF1 associates with TβRII and inhibits TGF-β receptor signalling.** TGF-β can promote invasion and metastasis in advanced tumours[3]. Consistent with previous reports[14,15], we observed that breast cancer cells with high metastatic potential appeared to have elevated TβRII protein levels (Supplementary Fig. 1a). Upon TβRII depletion, we observed a marked reduction of both breast cancer and lung cancer metastasis in xenograft mouse models (Fig. 1a; Supplementary Fig. 1b). Cells isolated from the metastatic nodules of mice showed a gain in TβRII protein (but not messenger RNA) expression compared with their parental cells, suggesting that TβRII protein is stabilized during cancer metastasis (Fig. 1b). We therefore sought to identify the critical regulators of TβRII. Treatment with lysosome inhibitors, such as bafilomycin A1, $NH_4Cl$ or chloroquine (but not the proteasome inhibitors MG132 or lactacystin), led to TβRII accumulation (Fig. 1c), suggesting that TβRII is degraded via a lysosomal pathway. We therefore analysed proteins that co-immunoprecipitated specifically with FLAG-tagged TβRII in the presence of lysosome inhibitor using mass spectrometry (Fig. 1d). FAF1, with 12 unique peptides detected, was identified as the strongest binding partner (Fig. 1d and Supplementary Table 1; Supplementary Data 1). By using limiting amounts of TβRII antibody in immunoprecipitation, we pulled down equal amounts of endogenous TβRII and verified that FAF1 bound to endogenous TβRII in $NH_4Cl$-treated non-transfected cells (Fig. 1e). Furthermore, the TGF-β-induced $CAGA_{12}$-Luc SMAD-dependent response was inhibited by FAF1 ectopic

expression and was enhanced by the depletion of endogenous FAF1 (Fig. 1f). These data suggest that FAF1 inhibits TGF-β signalling by transiently binding to TβRII, which may result in TβRII instability.

To validate this hypothesis, we first examined whether FAF1 affects the immediate mediators of TGF-β receptor signalling. The levels of TGF-β-induced SMAD2 phosphorylation (P-SMAD2), SMAD2–SMAD4 complex formation, phosphorylated AKT (P-AKT) and phosphorylated p38 MAP kinase (P-p38) were elevated in FAF1-depleted parental MDA-MB-231 cells (Fig. 1g). These responses were severely inhibited by the ectopic expression of FAF1-WT, but were not affected by a FAF1 mutant lacking the ubiquitin regulatory X (UBX) domain (Fig. 1h). However, FAF1 did not affect the total levels of SMAD2, SMAD4, AKT or p38 MAP kinase. These data suggest that endogenous FAF1 is a critical antagonist for both TGF-β/SMAD and non-SMAD signalling.

High-grade tumours and/or their cancer-associated fibroblasts (CAFs) frequently express elevated levels of TGF-β, which correlates with poor prognosis in cancer patients[3,16]. Given the role of FAF1 in restricting TGF-β signalling activity, we investigated the possibility that loss of FAF1 might be a relevant prognostic factor in late-stage cancer. Oncomine[17] expression analysis revealed FAF1 downregulation in multiple human cancers in which TGF-β is pro-metastatic (Fig. 1i; Supplementary Fig. 2a). Using The Cancer Genome Atlas patient database, we observed that claudin-low breast cancer patients show significant under-expressed FAF1 (Supplementary Fig. 2b). In the same database, individuals with breast carcinomas exhibiting higher FAF1 expression had a longer life expectancy than those with tumours exhibiting lower FAF1 expression (Fig. 1j). These data suggest that FAF1 may play a tumour-suppressing role in multiple human cancer types by inhibiting tumour-promoting pathways, such as TGF-β signals.

**FAF1 promotes turnover of cell surface TβRII.** The fact that depleting endogenous FAF1 enhances the half-life of exogenous C-terminal HA-tagged TβRII suggests that TβRII protein stability is highly regulated by FAF1 (Fig. 2a). To confirm this possibility, we measured the stability of endogenous TβRII. Pulse-chase labelling experiments showed that endogenous TβRII exhibited a shortened half-life upon the ectopic expression of FAF1-WT, but not upon the expression of a FAF1 mutant lacking the UBX domain (Fig. 2b). Conversely, TβRII displayed a prolonged half-life in FAF1-depleted cells (Supplementary Fig. 2c). We therefore investigated whether FAF1 misexpression affects TβRII levels at the plasma membrane, where signalling is initiated. Biotin-labelled cell surface TβRII displayed a severely decreased half-life upon ectopic FAF1 expression, but not upon expression of a FAF1 mutant lacking the UBX domain (Fig. 2c). As expected, FAF1 depletion in cells led to higher cell surface TβRII levels and an apparent decrease in protein turnover (Fig. 2d). These findings suggest that FAF1 could promote TβRII instability by stimulating to TβRII turnover at the plasma membrane.

**FAF1 recruits VCP/βTRCP to target TβRII for polyubiquitylation.** FAF1 is not an E3 ubiquitin ligase but an adaptor for substrate ubiquitylation and degradation[18–20], and TβRII is regulated by ubiquitination[21]. Analysis of the FAF1 interactome by mass spectroscopy examination of the co-immunoprecipitants of Flag-tagged FAF1 revealed that FAF1 bound to valosin-containing protein (VCP) at a ratio near 1:1 (Fig. 3a,b; Supplementary Data 2). VCP has been shown to interact with the FAF1 UBX domain[18]. We found that this UBX domain is required for FAF1 to antagonize cell surface TβRII (Fig. 2c). This finding suggests that FAF1 might recruit VCP to regulate TβRII.

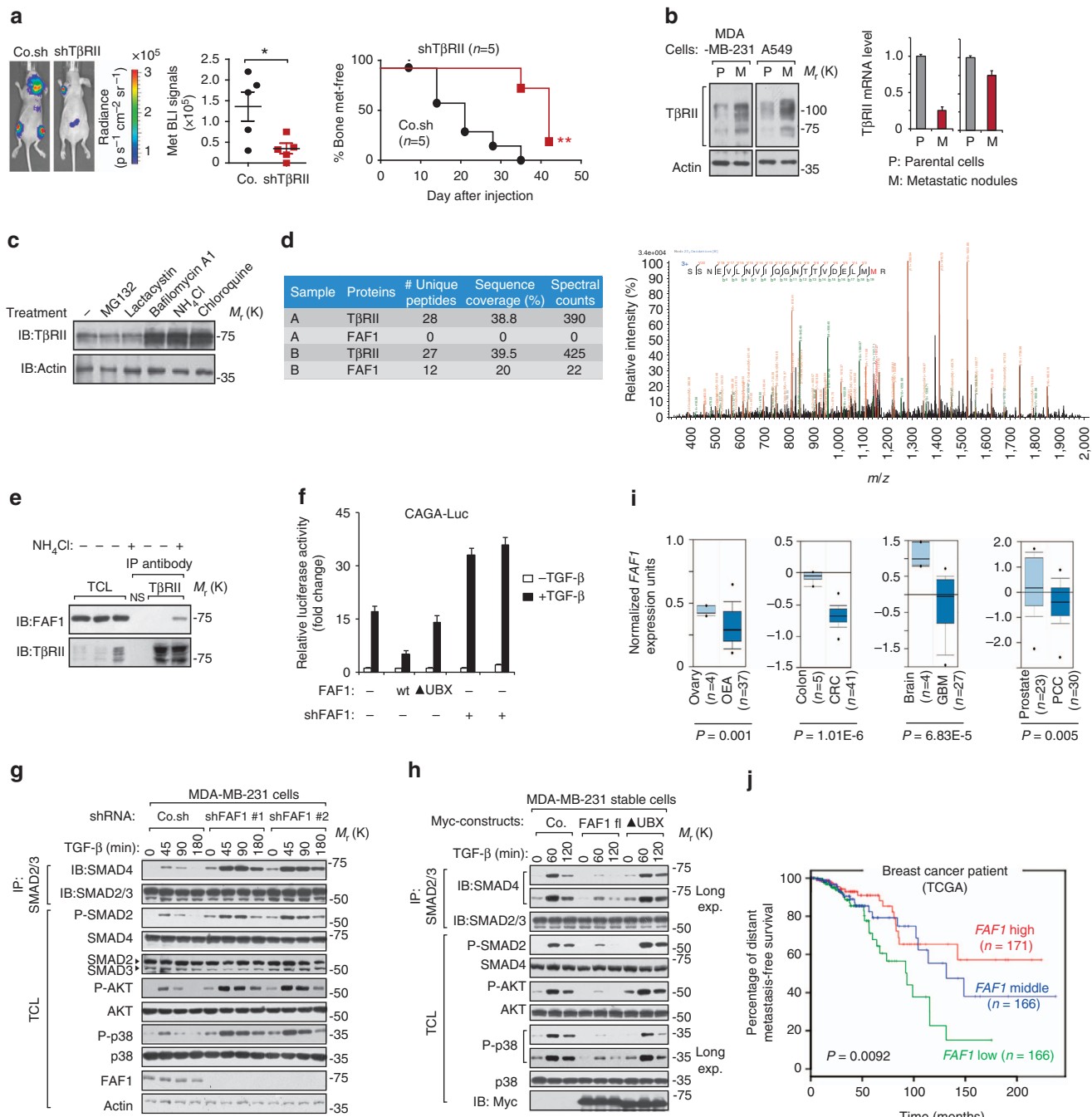

**Figure 1 | FAF1 specifically associates with TβRII and inhibits TGF-β signalling.** (**a**) Bioluminescent imaging (BLI) of representative mice from each group injected into the left heart ventricle with control (Co.) or MDA-MB-231 breast cancer cells stably depleted of TβRII (shTβRII). Images were captured at week 7. Dorsal images are shown. The BLI signal of every mouse in each experimental group is shown in the middle panel. The percentage of bone metastasis-free mice (right panel) in each experimental group is given. (**b**) Immunoblot (IB) analysis of TβRII protein levels in parental (P) and metastatic (M) variant of MDA-MB-231 and A549 cells (left panel). Quantitative PCR with reverse transcription analyses of *TβRII* in parental and metastatic cells are shown (right panel). Data are presented as the mean ± s.d. (**c**) MDA-MB-231 cells were treated with MG132 (25 μM), lactacystin (30 μM), bafilomycin A1 (1 μM), NH₄Cl (25 mM) or chloroquine (100 mM) for 5 h and collected for IB analysis. Actin IB is included as a loading control. (**d**) Mass spectrometry results of TβRII-Flag immunoprecipitants in the absence (sample A) and presence (sample B) of NH₄Cl (25 mM) identified FAF1 as a binding partner; exemplified peptide is shown in the right panel. (**e**) IB of the input and immunoprecipitants (IPs) derived from MCF10A-RAS cells treated with NH₄Cl (25 mM) as indicated; 5% of the total cell lysate was loaded as input. NS, non-specific antibody. (**f**) The CAGA₁₂-Luc SMAD-dependent transcriptional response in HEK293T cells transfected with FAF1 wt/UBX-deleted mutant (▲UBX) or shFAF1, as indicated, and treated with TGF-β (5 ng ml⁻¹) overnight. The data are presented as the mean ± s.d. of three independent sets of experiments. (**g,h**) The IBs of total cell lysate (TCL) and anti-SMAD2/3 immunoprecipitants derived from control and FAF1 stable depletion (shFAF1#1 and shFAF1#2) (**g**) or Myc-FAF1 fl (full length)/UBX-deleted mutant stably expressing (**h**) MDA-MB-231 cells treated with ligand at the indicated time points. (**i**) Oncomine box plots of the *FAF1* expression levels in multiple human advanced cancers versus normal tissues; *P* < 0.01. (**j**) Kaplan–Meier curves of The Cancer Genome Atlas database showing that the metastasis-free survival of individuals positively correlated with *FAF1* expression by the log-rank test (*P* < 0.01).

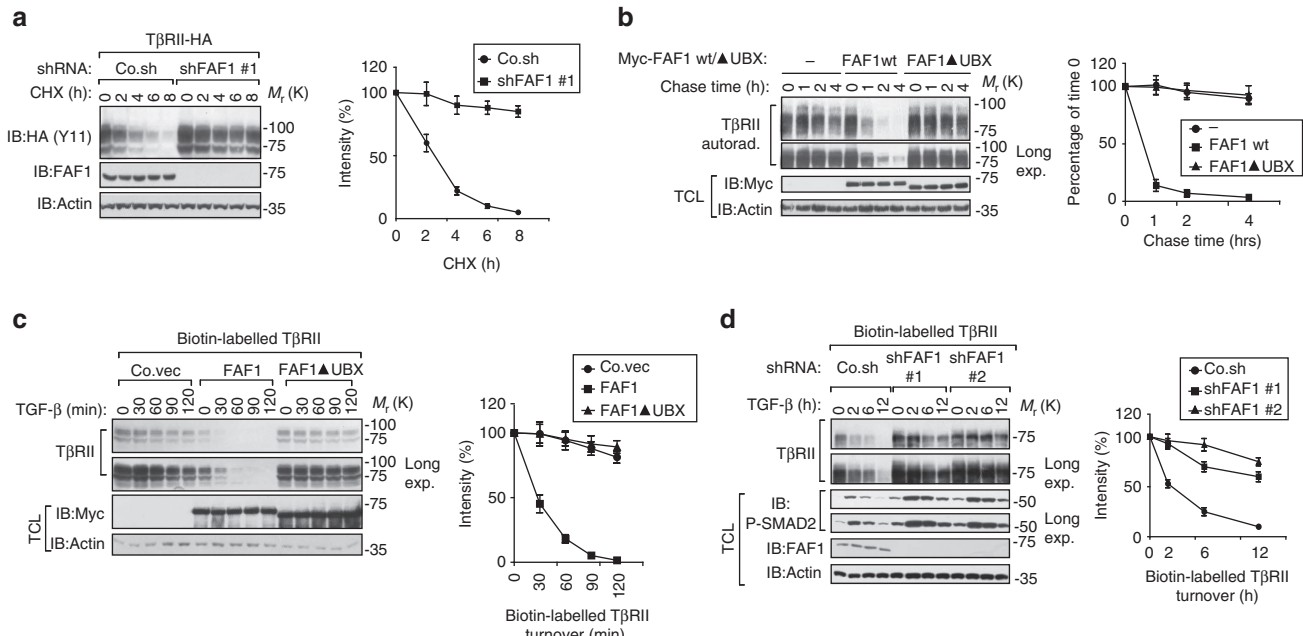

**Figure 2 | FAF1 promotes TβRII turnover at the cell surface. (a)** The immunoblot (IB) of cell lysate derived from HEK293T cells stably expressing TβRII-HA and depleted of FAF1 that were treated with CHX (20 μg ml$^{-1}$) at the indicated time points. The amount of protein after CHX treatment was expressed as a percentage of that present before treatment (time 0), and is shown in the right panel. The results are shown as the mean ± s.d. of three independent sets of experiments. **(b)** [$^{35}$S]-methionine labelling and pulse-chase analysis of TβRII in control and MDA-MB-231 cells stably overexpressing empty vector ( − ), Myc-FAF1 wt or FAF1▲UBX (UBX domain-deleted mutants). The amount of labeled protein precipitated after the chase was expressed as the percentage of that at the beginning of the chase (time 0) and is shown in the right panel. The results are given as the mean ± s.d. of three independent sets of experiments. **(c,d)** IB of biotinylated cell surface TβRII in MDA-MB-231 cells stably overexpressing FAF1 wt/FAF1 UBX-deleted mutant (▲UBX) **(c)** or stably depleted of endogenous FAF1 by two independent shRNA (shFAF1 #1 and shFAF1 #2) **(d)** and treated with TGF-β (5 ng ml$^{-1}$) at the indicated time points. Quantification of the band intensities is shown in the right panel. Band intensity was normalized to the $t = 0$ controls. The results are presented as the mean ± s.d. of three independent sets of experiments.

VCP is an evolutionarily conserved hexameric AAA (ATPases associated with diverse cellular activities) family member that can convert chemical energy from ATP hydrolysis into mechanical force to remodel protein complexes[22]. One such complex—VCP–ubiquitin fusion degradation 1 (UFD1)–nuclear protein localization 4 (NPL4)—plays a role in endoplasmic reticulum-associated degradation[22]. Other complexes, including VCP-p47 and VCP-UBXD1, have been proposed to catalyse proteasome-independent membrane trafficking, including membrane protein segregation[22]. Interestingly, all of the important VCP-binding partners were identified among FAF1 co-immunoprecipitants, including VAPA/VAPB, vesicle-associated membrane proteins that function in trafficking and membrane fusion, and the E3 ubiquitin ligases β-transducin repeat containing E3 ubiquitin protein ligase (βTRCP) and Cullin1/3 (Fig. 3a; Supplementary Data 2). All of these proteins were confirmed to bind FAF1 (Supplementary Fig. 3a).

As examined using a reporter assay, ectopic VCP expression severely impaired TGF-β/SMAD signalling, whereas knocking down endogenous VCP enhanced this response (Fig. 3c). In cells ectopically expressing VCP, TβRII cell surface localization was lost, and the protein appeared to localize in a dotted pattern in the cytosol (Fig. 3d). Furthermore, FAF1-mediated inhibition of TβRII expression was partially rescued by N2, N4-dibenzylquinazoline-2,4-diamine, an inhibitor of the ATPase activity of VCP (Supplementary Fig. 3b).

VCP–UFD1–NPL4 as well as the E3 ligases βTRCP and CULLIN1 were found to associate with TβRII in control cells but not in FAF1-depleted cells (Fig. 3e). The loss of FAF1 had no effect on the VCP–UFD1–NPL4 complex formation but led to a reduced association between VCP-TβRII and VCP-βTRCP

(Fig. 3f). These results indicated that FAF1 recruits βTRCP to the VCP complex and subsequently recruits VCP/βTRCP to TβRII. This finding might be relevant for cell surface TβRII, because we observed that FAF1 induced an increase in the levels of both VCP–UFD1–NPL4 and βTRCP in the membrane fraction (Supplementary Fig. 3c). To validate this hypothesis, we pulled down biotin-labelled cell surface proteins and found that FAF1-WT but not the UBX-deleted mutant increased plasma membrane-associated VCP–UFD1–NPL4/βTRCP/CULLIN1 and reduced the level of TβRII at the cell surface (Fig. 3g). In the same assay, FAF1 depletion reduced cell surface-bound VCP–UFD1–NPL4/βTRCP/CULLIN1 and increased the amount of TβRII at the plasma membrane (Fig. 3h).

βTRCP is a F-box protein[23]. SGT1A, a homolog of S-phase kinase-associated protein 1 (Skp1), was also identified as a FAF1-binding partner (Fig. 3a; Supplementary Data 2). Thus, FAF1 actually pulled down a SKP1-cullin1-F-box protein E3 ligase complex in which the substrate-recognition subunit βTRCP plays an essential role[24,25]. In the ubiquitylation assay, FAF1 promoted TβRII polyubiquitylation, an effect that was inhibited upon βTRCP depletion (Fig. 3i). Depletion of FAF1 together with βTRCP therefore greatly decreased TβRII polyubiquitylation (Fig. 3j). These results indicate that TβRII polyubiquitylation is primarily accomplished by the FAF1/βTRCP complex. Furthermore, ectopic expression of VCP also promoted TβRII polyubiquitylation and TβRII turnover on the cell surface—effects that were dependent on endogenous FAF1 expression (Fig. 3k). In summary, TβRII is primarily targeted for polyubiquitylation and turnover in the lysosome by the VCP and βTRCP E3 ubiquitin ligase complex, which is recruited by FAF1 (Fig. 3l).

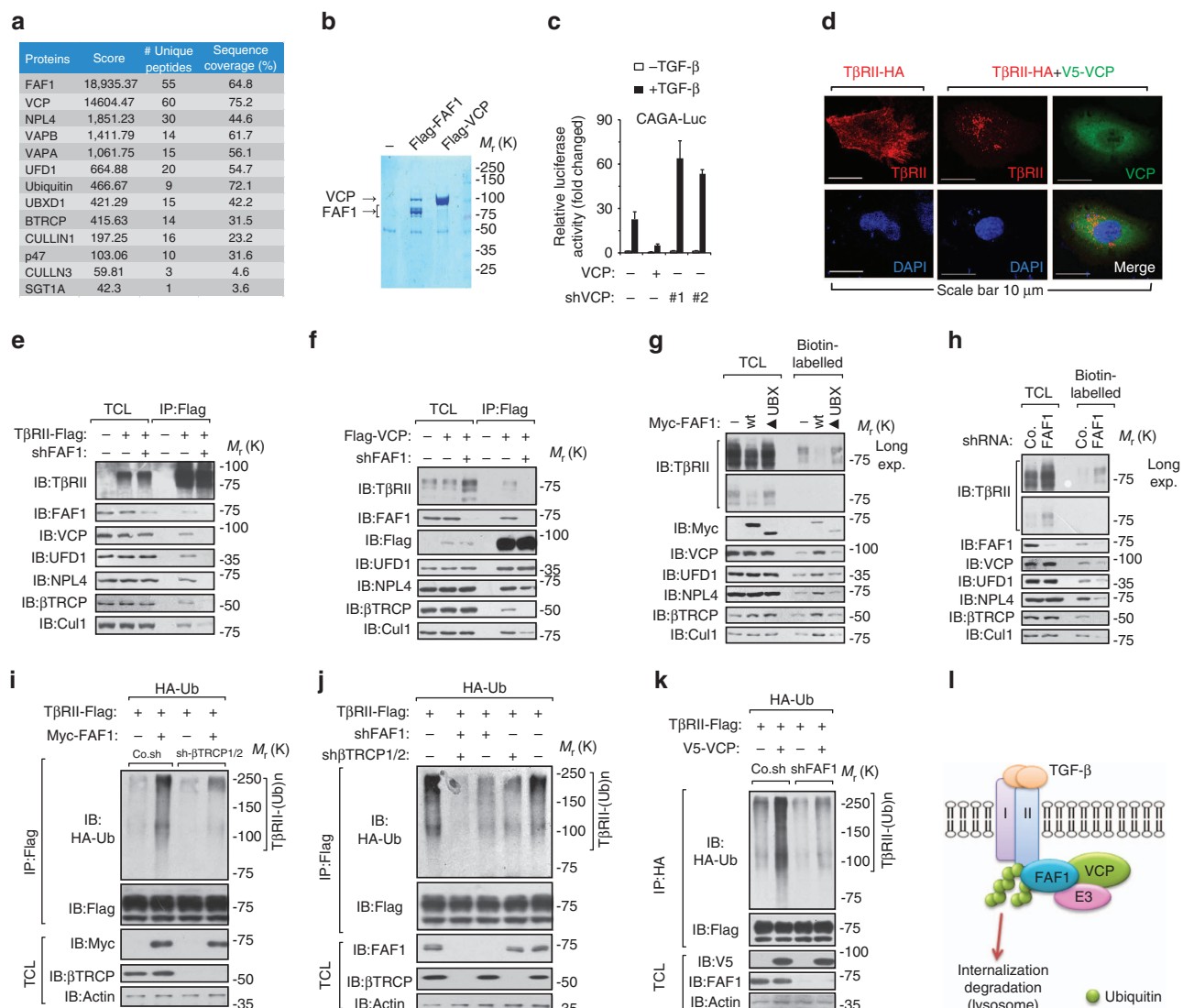

**Figure 3 | FAF1 recruits VCP/βTRCP to target TβRII for polyubiquitination.** (**a**) Mass spectrometry studies of anti-Flag-FAF1 immunoprecipitants revealed FAF1-binding partners. (**b**) FAF1 bound to VCP at a ratio close to 1:1. (**c**) The CAGA$_{12}$-Luc SMAD-dependent transcriptional response of HEK293T cells expressing VCP or VCP shRNA (#1 and #2) and treated with TGF-β (2.5 ng ml$^{-1}$) overnight is shown. The data are presented as the mean ± s.d. of three independent sets of experiments. (**d**) Immunofluorescence and 4, 6-diamidino-2-phenylindole (DAPI) staining of HeLa cells transfected with TβRII-HA or TβRII-HA with V5-VCP plasmids. Scale bar, 10 μm. (**e,f**) HEK293T cells were transfected with TβRII-Flag (**e**) or Flag-VCP (**f**) and with FAF1 depletion or not, as indicated. Cells were collected for immunoprecipitation (IP) and immunoblot (IB) analysis; 5% of the total cell lysate was loaded as an input. (**g,h**) MDA-MB-231 cells expressing Myc-FAF1 wt/UBX-deletion mutant (▲UBX) (**g**) or infected with FAF1 shRNA as indicated (**h**) were harvested for biotinylation and immunoblot (IB) analysis. Co., control non-targeting shRNA. (**i**) HA-Ub stably expressing HEK293T cells were transfected with TβRII-Flag along with Myc-FAF1 and were infected with control (Co.sh) or βTRCP shRNA lentivirus as indicated. The cells were then collected for IP and IB analysis. (**j**) HA-Ub stably expressing HEK293T cells were transfected with TβRII-Flag and infected with βTRCP1/2 shRNA and/or FAF1 shRNA lentivirus as indicated. Cells were then collected for IP and IB analysis. (**k**) HA-Ub stably expressed HEK293T cells were transfected with TβRII-Flag and VCP, and were infected with FAF1 shRNA lentivirus as indicated. Cells were then collected for IP and IB analysis. (**l**) Working model of how FAF1 mediates TβRII polyubiquitylation and turnover via lysosomal mediated degradation.

**FAF1 correlates with good prognosis in breast cancer patients.** Epithelial-like breast cancer cells can undergo epithelial-to-mesenchymal transition (EMT), upon which they become more migratory, invasive and metastatic[26–28]. The upregulation of mesenchymal markers N-cadherin, fibronectin, smooth muscle actin and vimentin, and the downregulation of epithelial marker E-cadherin are typical characteristics of EMT. Gene set enrichment analysis (GSEA) of the NKI breast cancer patients data set[29] demonstrated that gene signatures representing a good prognosis or less invasiveness of the tumour were significantly enriched in patients with higher levels of *FAF1* expression

($n = 103$) compared to those with lower expression of this gene ($n = 104$), strongly suggesting that FAF1 correlates with a good prognosis (Fig. 4a).

Using immunofluorescence, we found that FAF1-expressing cells exhibited an epithelial-like phenotype similar to those of control cells and cells treated with the TβRI kinase inhibitor SB431542, whereas FAF1-depleted cells exhibited mesenchymal-like characteristics. The latter phenotype was partially inhibited by SB431542 (Supplementary Fig. 4a). Upon ectopic FAF1 expression, the changes in EMT marker expression induced by TGF-β were attenuated, whereas FAF1 depletion had the reverse

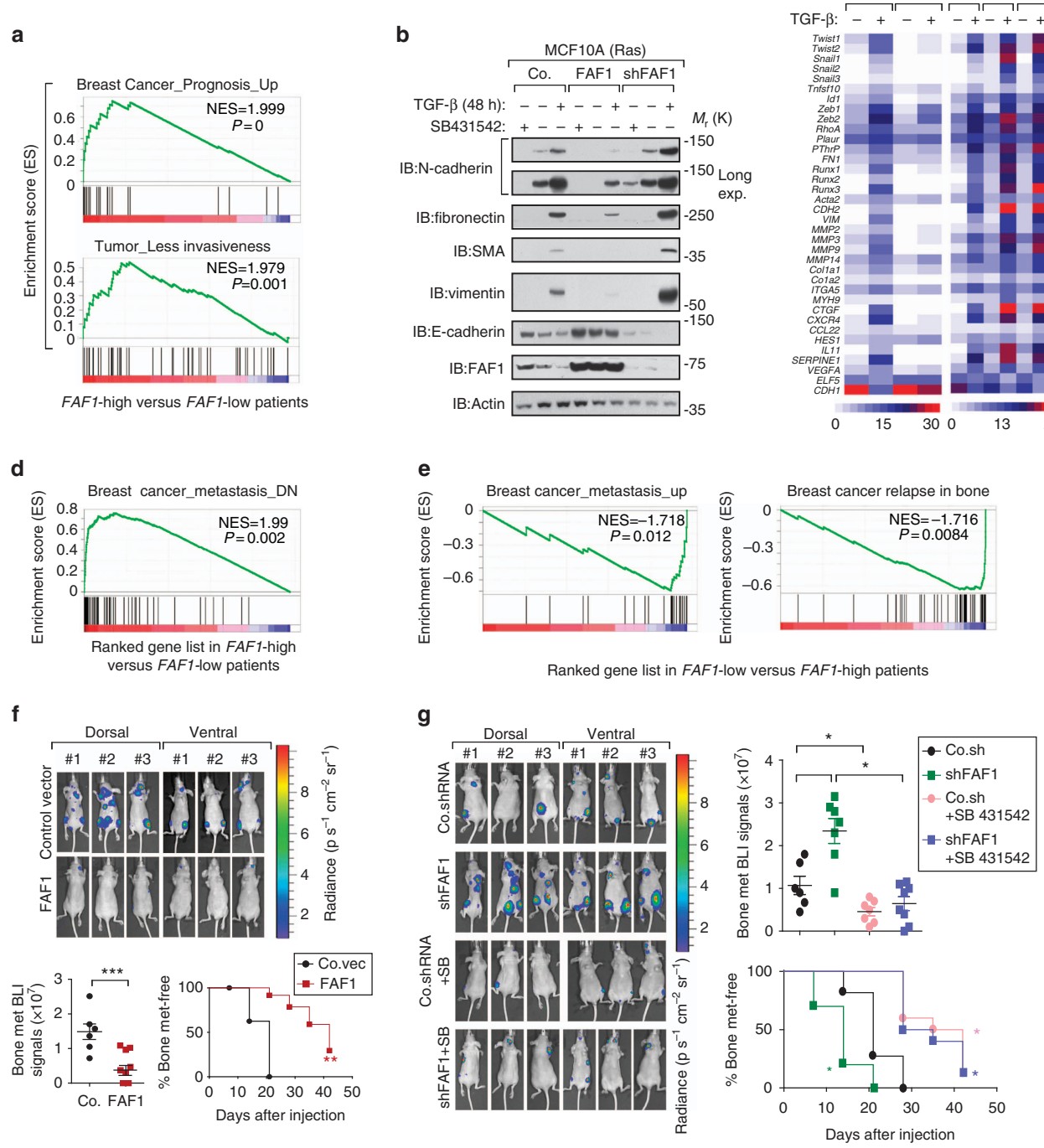

**Figure 4 | FAF1 inhibits bone metastasis in transplantable mouse model of metastasis and correlates with good prognosis in breast cancer patients.**
(**a**) Pre-ranked gene set enrichment analysis (GSEA) in *FAF1*-high versus *FAF1*-low patients. The consensus signature of the genes correlated with good prognosis (upper panel) or lower invasiveness (lower panel) was enriched in *FAF1*-high compared with *FAF1*-low patients. (**b**) Immunoblot (IB) of control and MCF10A-RAS cells stably expressing FAF1 or depleted of endogenous FAF1 (shFAF1) and treated with TGF-β (5 ng ml$^{-1}$) and SB431542 (10 μM) as indicated for 48 h. Epithelial-to-mesenchymal transition (EMT) marker proteins were analysed as indicated. (**c**) Heat map of TGF-β target and EMT-related genes in control or MCF10A-RAS cells stably expressing FAF1 or depleted of FAF1 (#1 and #2) and treated with or without TGF-β (1 ng ml$^{-1}$) for 8 h. (**d**) Consensus signature of genes significantly and negatively correlated with breast cancer metastasis are enriched in *FAF1*-high versus *FAF1*-low patients, as shown by pre-ranked GSEA. (**e**) Consensus signature of genes significantly and positively correlated with breast cancer metastasis (left panel), especially bone metastasis (right panel), are enriched in *FAF1*-low versus *FAF1*-high patients, as shown by GSEA (right panel). (**f**) Bioluminescent imaging (BLI) of three representative mice from each group at week 6 following injection with control or MDA-MB-231 cells stably expressing FAF1. Both ventral and dorsal images are shown (upper panel). The BLI signal of every mouse (lower left panel) and the percentage of bone metastasis-free mice (lower right panel) in each experimental group were followed over time. (**g**) The BLI of three representative mice from each group at week 6 following injection with control or MDA-MB-231 cells stably depleted of FAF1 and pretreated with or without SB431542 (SB, 10 μM) overnight. Both ventral and dorsal images are shown (left panel). The BLI signal of every mouse (right upper panel) and the percentage of bone metastasis-free mice (right lower panel) in each experimental group are shown over time. Data are presented as the mean ± s.d.

effect (Fig. 4b). To evaluate whether FAF1 regulates global EMT-related changes, we performed gene expression profiling of RAS-transformed MCF10A cells. PCR array analyses indicated that the loss of FAF1 led to several molecular features of EMT, including the upregulation of key transcriptional inducers such as *Twist1*, *Twist2*, *Snail1*, *Snail2*, *Snail3*, *Zeb1* and *Zeb2*. Importantly, all of these genes were suppressed by an increase of FAF1 either in the basal or TGF-β-induced level (Fig. 4c), indicating that FAF1 could promote mesenchymal-to-epithelial transition. In line with this finding, FAF1 attenuated TGF-β-induced migration in two-dimensional transwell assays (Supplementary Fig. 4b) as well as invasion in three-dimensional spheroid-collagen culture systems (Supplementary Fig. 4c). In addition, knockdown of endogenous FAF1 promoted breast cancer migration and invasion (Supplementary Fig. 4b,d). Moreover, GSEA analysis showed that genes correlating with decreased metastasis were significantly enriched in *FAF1*-high patients (Fig. 4d), whereas gene signatures linked to increased metastasis, especially in the bone, were significantly enriched in *FAF1*-low patients (Fig. 4e). We therefore directly investigated whether FAF1 suppresses bone metastasis in a transplant mouse model.

We used the highly bone metastatic subline MDA-MB-231 to evaluate the effect of FAF1 on metastasis[30–32]. Mice that had been intracardially injected with control MDA-MB-231 cells began to develop detectable bone metastases after 35 days, and the number of metastatic nodules and the area covered by them expanded in the following 2 weeks. Mice injected with FAF1-expressing cells developed fewer bone metastases and had significantly longer bone metastases-free survival periods (Fig. 4f). Furthermore, the intracardial injection of cells expressing FAF1 short hairpin RNA (shRNA) revealed potentiated bone colonization of circulating MDA-MB-231 cells, which was partially blocked by pretreatment of the cells with SB431542 (Fig. 4g). This finding confirms that FAF1 inhibits breast cancer bone metastasis, at least in part, via the suppression of TGF-β receptor signalling.

**FAF1 depletion promotes lung metastasis in the MMTV-PyMT mouse model.** To determine the *in vivo* physiological functions of FAF1, we generated FAF1-knockout mice by utilizing gene-targeted ES cells in which exon 3 of the FAF1 gene had been replaced with the neomycin-resistance gene to generate a frame shift in the remaining FAF1 sequence (Fig. 5a; Supplementary Fig. 5). FAF1 protein expression was greatly reduced in FAF1 heterozygous embryos and was absent in FAF1 homozygous embryos (Fig. 5b), confirming gene knockout. FAF1$^{-/-}$ female mice were viable and fertile and displayed no obvious abnormalities when monitored for up to 1 year. In FAF1$^{+/-}$ and FAF1$^{-/-}$ mouse embryonic fibroblast cells, we observed stabilized cell surface localization of TβRII, which led to a more sensitive TGF-β response as indicated by increased TGF-β-induced phosphorylation of both SMAD2 and AKT (Fig. 5c). The expression of downstream TGF-β target genes, such as plasminogen activator inhibitor 1 (*PAI1*), interleukin 11 (*IL11*) and N-Cadherin (*CDH2*), were also elevated in FAF1$^{+/-}$ and FAF1$^{-/-}$ mouse embryonic fibroblasts (Fig. 5d).

To complement our xenograft metastasis models, we examined the effects of FAF1 on metastasis using the well-established MMTV-PyMT transgenic mouse model. These mice develop luminal adenocarcinoma with a high incidence of lung metastasis, which relies on AKT activity[33,34]. Similar to the results obtained from the xenograft models, we observed that FAF1$^{-/-}$/PyMT and FAF1$^{+/-}$/PyMT mice exhibited an increase in the incidence of lung metastasis (Fig. 5e), as well as a significant increase in the number of lung metastasis nodules (Fig. 5f) and greater lung lesion surface area (Fig. 5g,h) compared with their FAF1$^{+/+}$/PyMT counterparts. Furthermore, the level of TβRII expression

was strongly increased in FAF1$^{-/-}$/PyMT lesions compared with FAF1$^{+/+}$/PyMT metastasis lesions (Fig. 5h). Finally, these results were confirmed by independent experiments in which FAF1-overexpressing primary tumour cells derived from MMTV-PyMT transgenic mice (Fig. 5i) exhibited a significant decrease in lung metastasis without detectable changes in primary tumour growth following mammary fat pad injection in Friend Virus B (FVB) mice (Fig. 5j–m). Overall, our data from transgenic mice experiments complement our findings from cell line models and clinical breast cancer samples, strongly supporting the role of FAF1 as a metastasis-suppressing gene via the direct targeting of TβRII.

**AKT phosphorylates FAF1 to inhibit its membrane localization.** In FAF1-expressing epithelial-like cells, the initiation of an oncogenic TGF-β signal requires the dissociation of FAF1 from TβRII. We next investigated how this process is achieved. In our mass spectrometry analysis, we found that FAF1 was phosphorylated on residues Ser 270, Ser 320 and Ser 582 (Supplementary Fig. 6a; Supplementary Table 2). Ser 582 is an AKT consensus RxRxxS(T) phosphorylation motif conserved in FAF1 orthologs (Fig. 6a,b); the other conserved residue, Ser 426, was not phosphorylated (Supplementary Fig. 6b). FAF1 was associated with AKT *in vitro* and *in vivo* (Supplementary Fig. 6c,d). We used a phospho-specific antibody that recognizes the optimal AKT phosphorylation consensus motif and found that endogenous FAF1 was indeed phosphorylated by AKT; this phosphorylation was elevated by an activated allele of AKT (Myr-AKT1) and was blocked by LY294002, a selective inhibitor of PI3K (Fig. 6c). Using *in vitro* and *in vivo* assays, we found that AKT phosphorylated FAF1 (Fig. 6d; Supplementary Fig. 6e). Phosphorylation was not detected for the FAF1 S582A mutant but was present in FAF1 protein with other sites mutated (Fig. 6d). The reactivity of FAF1 with the p-AKT substrate antibody was reversed when the cell lysates were incubated with lambda phosphatase (Supplementary Fig. 6f). These data strongly suggest that AKT specifically phosphorylates FAF1 at Ser 582. Under physiological conditions, both insulin-like growth factor (IGF)-1 and TGF-β-induced phosphorylation of endogenous FAF1 were detected by the AKT substrate antibody and were abolished when the cells were treated with LY294002 (Fig. 6e), indicating that FAF1 is regulated by various cytokines and oncogenic stimuli that activate AKT.

The phosphorylation of certain AKT substrates controls their subcellular localization[35]. Cell fractionation analysis showed that activated Myr-AKT inhibits the membrane localization of FAF1-WT, but not of the phospho-resistant FAF1-S582A (Fig. 6f). Compared with FAF1-WT, the basal level of FAF1-S582A in the membrane fraction was increased, whereas the phospho-mimetic FAF1-S582D was undetectable (Supplementary Fig. 7a). Upon LY294002 treatment or the expression of activated AKT, endogenous FAF1 showed increased or reduced localization, respectively, at the membrane, leaving the FAF1 level unchanged in both the cytoplasmic and nuclear fractions (Fig. 6g). Similarly, TGF-β reduced the membrane association of FAF1, which was rescued by the selective inhibition of either PI3K (LY294002 or BKM120) or AKT (GDC0068 or MK2206), suggesting that the effect was dependent on PI3K/AKT activity (Fig. 6h; Supplementary Fig. 7b). Therefore, Ser 582 phosphorylation by AKT inhibited the membrane localization of FAF1. We further hypothesized that this phosphorylation might also affect the capacity of FAF1 to recruit the VCP/E3 complex (Fig. 6i).

**AKT phosphorylates FAF1 thus disrupts FAF1-VCP/βTRCP complex.** Ser 582 is located in the FAF1 UBX domain, a region that is required for FAF1's association with VCP and βTRCP[18,20].

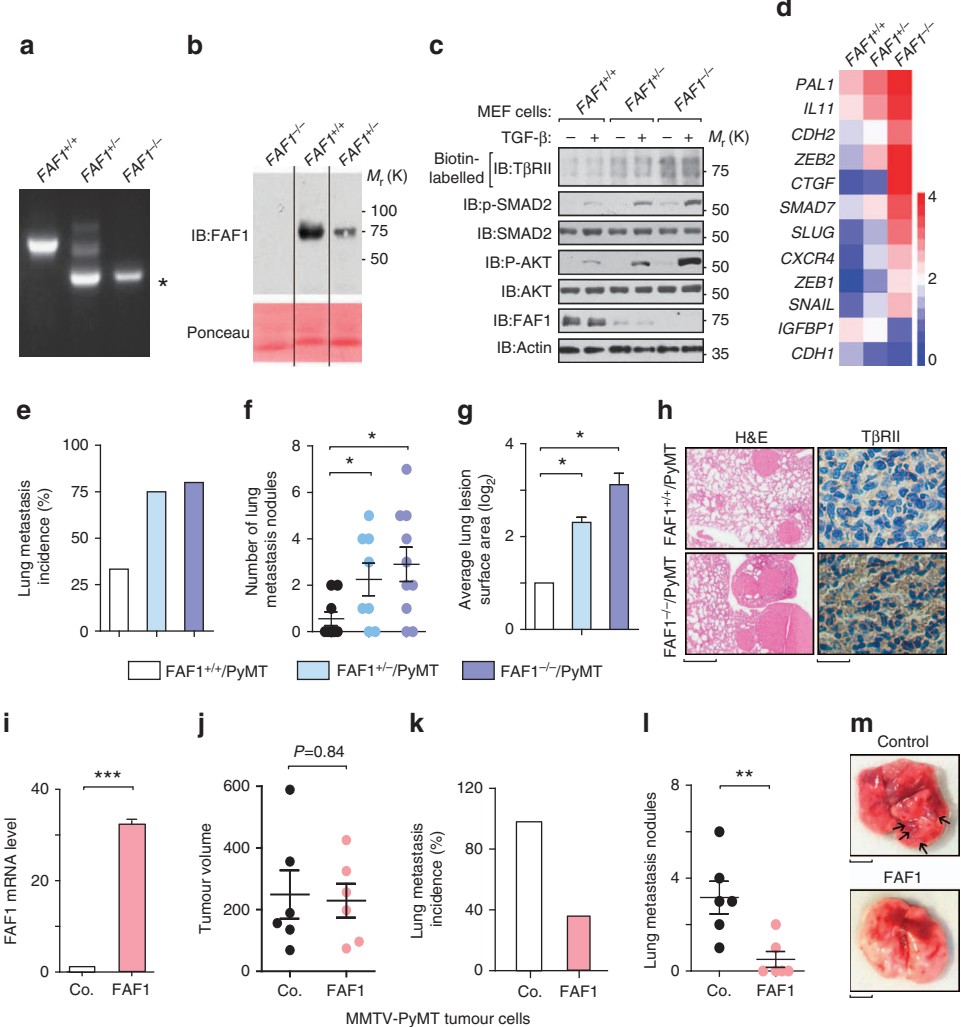

**Figure 5 | Loss of *Faf1* in mice results in the accumulation of TβRII and promotes lung metastasis in the MMTV-PyMT transgenic mouse model.**
(**a**) Genotyping of the generated FAF1$^{+/+}$, FAF1$^{+/-}$ and FAF1$^{-/-}$ mouse lines. PCR analysis of genomic DNA obtained from the tails of E13.5 mouse embryos. The asterisk indicates a FAF1 knockout-specific band. (**b**) IB of FAF1 protein derived from mouse embryonic fibroblasts (MEFs) from E13.5 FAF1$^{+/+}$, FAF1$^{+/-}$ and FAF1$^{-/-}$ embryos. (**c**) FAF1$^{+/+}$, FAF1$^{+/-}$ and FAF1$^{-/-}$ primary MEFs from E13.5 embryos were treated with TGF-β (5 ng ml$^{-1}$) for 45 min and then collected for biotinylation and IB analysis for cell surface TβRII expression. Expression levels of (phosphorylated) SMAD2 and AKT are also shown by IB with indicated antibodies. (**d**) FAF1$^{+/+}$, FAF1$^{+/-}$ and FAF1$^{-/-}$ primary MEFs from E13.5 embryos were collected for quantitative PCR with reverse transcription (qRT–PCR) analysis of TGF-β-targeted and invasion-related genes. Relative mRNA levels are shown as a heat map. (**e–g**) The incidence of spontaneous lung metastasis (**e**), the number of lung metastasis lesions (**f**) and the average lung lesion surface area (**g**) (arbitrary units based on pixel quantification from digital images, the data represented are shown as the mean ± s.d.) from FAF1$^{+/+}$/MMTV-PyMT$^{+}$ (*n* = 9), FAF1$^{+/-}$/MMTV-PyMT$^{+}$ (*n* = 8) and FAF1$^{-/-}$/MMTV-PyMT$^{+}$ animals (*n* = 10). *P < 0.05 by Mann–Whitney U-test in **f** and Student's t-test in **g**. (**h**) TβRII staining in lung lesions collected from FAF1$^{+/+}$/MMTV-PyMT$^{+}$ or FAF1$^{-/-}$/MMTV-PyMT$^{+}$ mice. Scale bars, 1 mm and 20 μm for lung haematoxylin and eosin (H&E) and TβRII immunohistochemistry images, respectively. (**i–m**) Overexpression of FAF1 in MMTV-PyMT tumour cells inhibits lung metastasis. (**i**) qRT–PCR analysis of FAF1 expression in control or FAF1-overexpressing MMTV-PyMT primary tumour cells. Experiments were performed three times, each with qRT–PCR in technical duplicates. The data are presented as the mean ± s.d. (**j**) Volumes of primary mammary fat pad tumours; *n* = 6 per experimental group. P value computed by Mann–Whitney U-test. (**k,l**) The incidence of spontaneous lung metastasis (**k**) and the number of lung metastasis lesions (**l**) in control and FAF1-overexpressing primary tumours; *n* = 6 per experimental group, **P < 0.01 by Mann–Whitney U-test in **l**. (**m**) Representative lung nodules from mice injected with control and FAF1-overexpressing MMTV-PyMT tumour cells. Black arrowheads indicate lung metastasis nodules. Scale bar, 2 mm.

This fact suggests that the binding of FAF1 to VCP might be influenced by AKT. In Myr-AKT-expressing cells, FAF1 barely bound to TβRII and inefficiently recruited VCP-UFD1-NPL4 or βTRCP/CULLIN1 (Fig. 7a). To verify whether this finding is due to Ser 582 phosphorylation, we compared interactions using FAF1-WT/S582A/S582D constructs. FAF1-S582A showed a higher binding affinity for TβRII, VCP-UFD1-NPL4 and βTRCP/Cullin1, whereas FAF1-S582D exhibited decreased interactions with these proteins (Fig. 7b). In line with this

finding, FAF1-S582A potentiated TβRII ubiquitylation more efficiently (Fig. 7c). In cells expressing FAF1-WT/S582A/S582D mutants, we isolated cell surface biotin-labelled proteins and found that FAF1-S582A significantly increased VCP-UFD1-NPL4 and βTRCP/Cullin1 in the cell membrane fraction, thereby reducing the cell surface expression of TβRII (Fig. 7d). This suggests that the negative effects of FAF1 on TβRII could be directly inhibited by the AKT-mediated FAF1 phosphorylation of Ser 582.

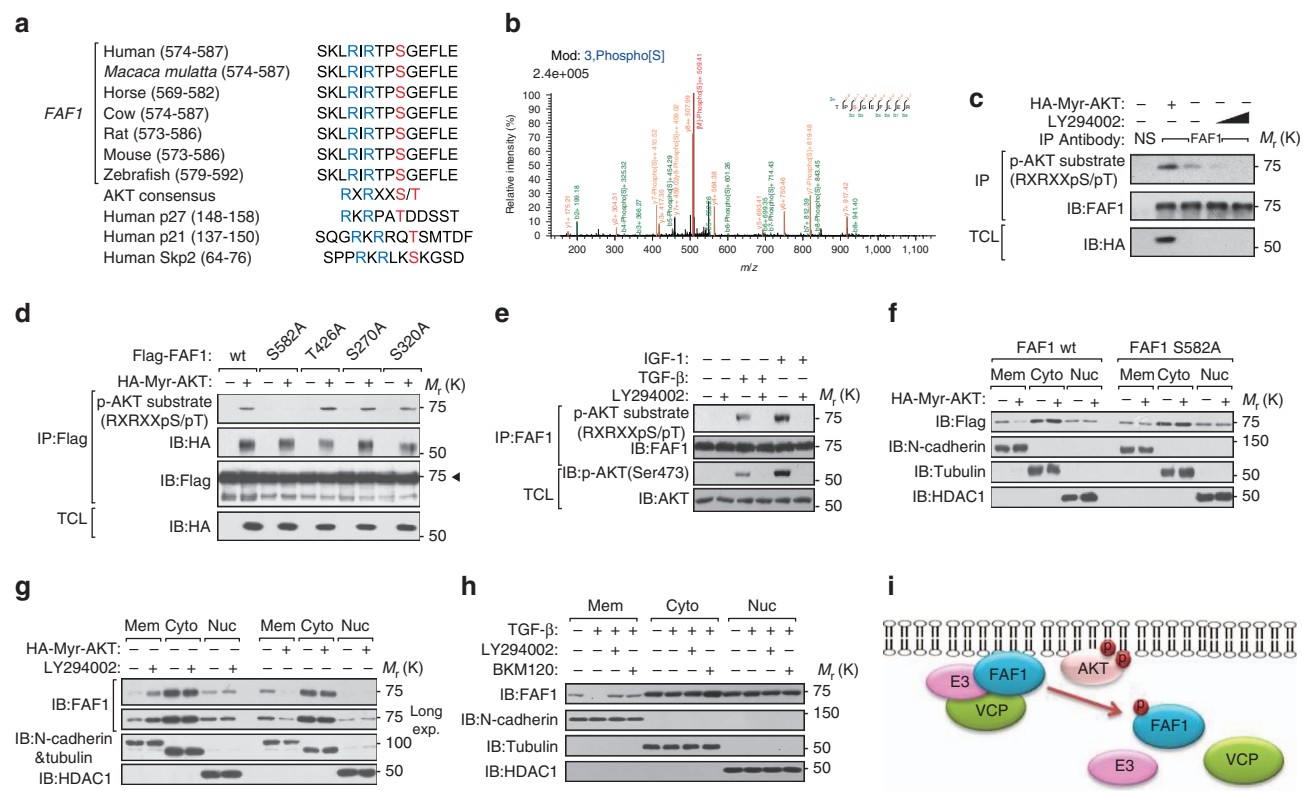

**Figure 6 | AKT phosphorylates FAF1 at Ser 582 and inhibits its membrane localization.** (**a**) Sequence alignment of the AKT phosphorylation site within FAF1 orthologs from different species and the known AKT-phosphorylating proteins p27, p21 and Skp2. (**b**) AKT-phosphorylated FAF1 peptide identified by mass spectrometry analysis. (**c**) Immunoblot (IB) of total cell lysate (TCL) and immunoprecipitants derived from HeLa cells transfected with Myr-HA-AKT and treated with increasing doses of LY294002 (25 and 50 μM) for 12 h as indicated. NS, non-specific antibody. (**d**) HEK293T cells were transfected with Flag-FAF1 wt or S582A, T426A, S270A or S320A mutants along with Myr-HA-AKT expression plasmids as indicated. Cells were then collected for immunoprecipitation (IP) and IB analysis. (**e**) IB analysis of TCL and anti-FAF1 immunoprecipitates derived from serum-starved HeLa cells treated with IGF-1 (200 ng ml$^{-1}$), TGF-β (5 ng ml$^{-1}$) and LY294002 (50 μM) as indicated for 8 h. (**f**) HeLa cells stably overexpressing FAF1 wt or FAF1 S582A mutant constructs were transfected with or without HA-Myr-AKT as indicated. Cells were then collected for membrane (Mem), cytoplasm (Cyto) and nuclear (Nuc) extraction and IB analysis. (**g**) IB of Mem, Cyto and Nuc extraction derived from HeLa cells transfected with or without HA-Myr-Akt and treated with LY294002 (50 μM for 12 h) as indicated. (**h**) HeLa cells were treated with TGF-β (5 ng ml$^{-1}$) and selective PI3K inhibitors LY294002 (50 μM) or BKM120 (1 μM) as indicated for 8 h. Cells were then collected for Mem, Cyto and Nuc extraction and IB analysis. (**i**) Hypothetical working model of AKT-mediated FAF1 phosphorylation: phosphorylated FAF1 does not efficiently attach to the cell surface or associate with the VCP/E3 complex.

These data suggest that the initial activation of AKT (for example, by oncogenic mutation or the activation of receptor tyrosine kinases) dissociates FAF1 from TβRII. TβRII may in turn lead to the further enhancement of SMAD and non-SMAD AKT activation downstream of TβRII. This process is a self-enforcing loop. If this mechanism indeed occurs during cancer progression, PI3K/AKT activity should correlate with TβRII levels in malignant cells. Indeed, blocking PI3K/AKT activity enhanced TGF-β-induced and FAF1-dependent TβRII ubiquitylation (Fig. 7e). AKT inhibited TβRII polyubiquitylation induced by FAF1-WT but had no effect in the case of FAF1-S582A (Supplementary Fig. 7c). TβRII was reduced to a low level by LY294002 in bone metastatic MDA-MB-231 cells, which impaired SMAD phosphorylation by TGF-β (Fig. 7f; Supplementary Fig. 7d). This process is apparently related to FAF1 recruitment of VCP; first, LY294002-induced loss of TβRII was rescued by N2, N4-dibenzylquinazoline-2,4-diamine, a selective VCP inhibitor (Fig. 7f). Second, LY294002-induced decrease of cell surface TβRII was not observed in FAF1-depleted cells (Fig. 7g).

To directly implicate FAF1 and its regulation by AKT-mediated phosphorylation in metastasis, we used a doxycycline-regulated promoter to express FAF1 wt or the phosphorylation-resistant S582A mutant in MDA-MB-231 cells

4 weeks after intracardial injection (Fig. 7h). The immediate expression of FAF1 wt significantly delayed metastatic outgrowth, whereas the expression of FAF1 S582A nearly abolished it (Fig. 7h). This result suggests that the induction of FAF1 expression could inhibit metastasis, but FAF1 protein was functionally suppressed by endogenous AKT kinase. In summary, AKT antagonizes FAF1 to maintain TβRII stability in metastatic (malignant) breast cancer cells (Fig. 7i).

The non-SMAD signals downstream of TβRII incorporates AKT, leading to the positive amplification loop of AKT-TGF-β signalling (Fig. 7i).

**Loss of FAF1 correlates with gain of TβRII function in patients.** Our findings suggest that pre-activated AKT phosphorylates and therefore antagonizes FAF1, which subsequently strengthens cell surface TβRII and results in the reinforced co-activation of SMAD and non-SMAD signalling (for example, AKT), which promote metastasis. To verify this model, we used an orthotopic mouse model. Pre-treatment with epidermal growth factor (EGF) at a low dose promoted the lung metastasis of 4T1 cells without significantly altering primary tumour growth under the nipple; this effect on metastasis was further promoted by FAF1 depletion (Fig. 8a,b).

To determine the clinical relevance and validity of our findings, we examined the expression of TβRII, P-SMAD2 and P-AKT 473, and their relationship with FAF1 in patient-derived tissue samples. We performed immunohistochemistry analysis on invasive breast carcinomas (110 cases), with tumour-adjacent normal breast tissue samples (11 cases) as controls. TβRII, P-SMAD2 and P-AKT 473 levels were significantly higher in breast carcinomas, whereas the FAF1 level was significantly lower in cancer tissues compared with normal breast tissues (Fig. 8c). Consistent with its role in mitigating TβRII protein levels, we

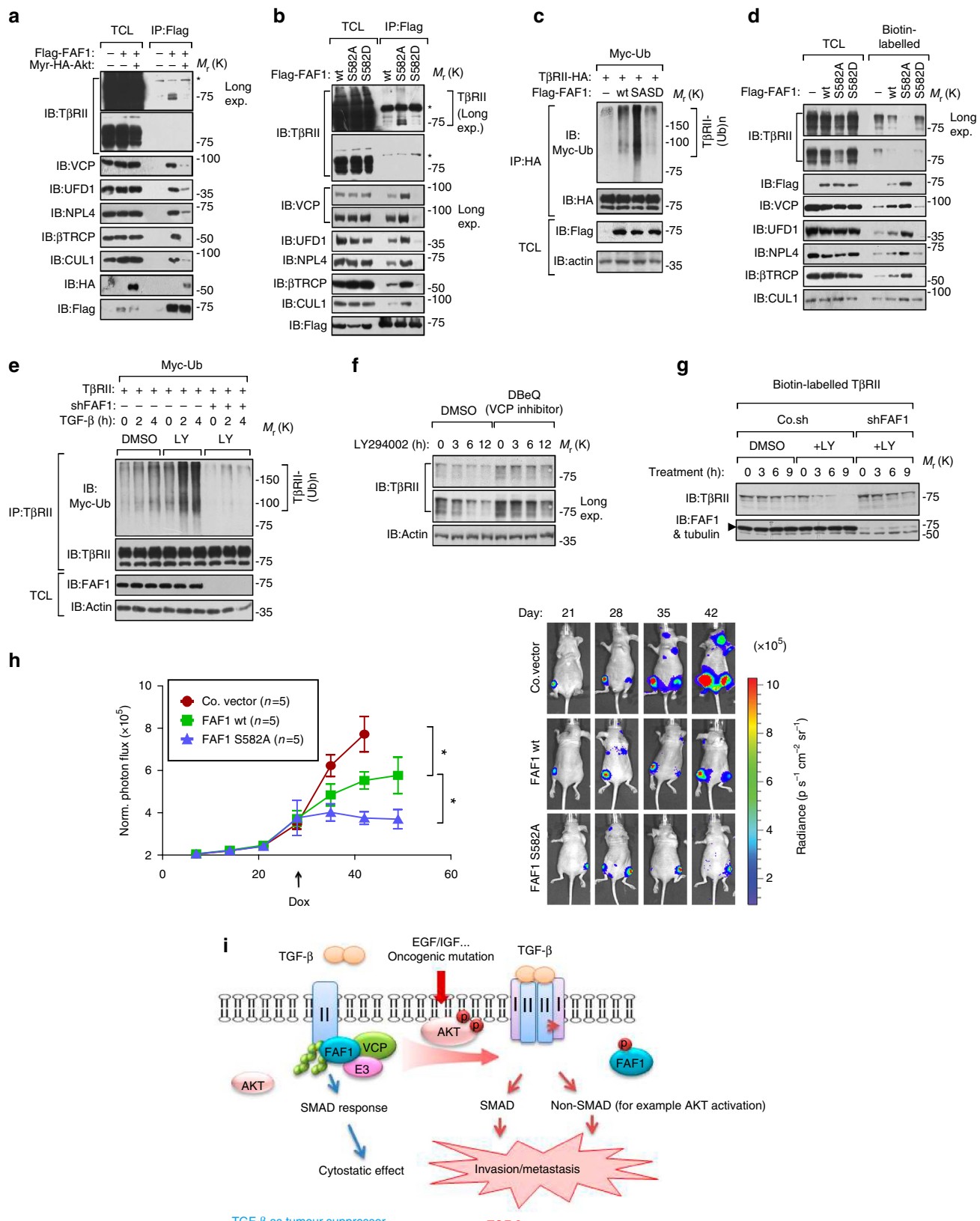

observed a significant inverse correlation between high FAF1 level and high levels of TβRII (Pearson $\chi^2 = 38.03$, $P = 3.5E - 08$), P-SMAD2 (Pearson $\chi^2 = 7.85$, $P = 0.001$) and P-AKT 473 (Pearson $\chi^2 = 1.88$, $P = 0.05$; Fig. 8d,e; Supplementary Fig. 8a,b). As shown in Fig. 8d, representative staining of the serial sections showed that the relative low FAF1 staining correlates with high levels of TβRII, P-SMAD2 and P-AKT 473, which were all found to be low in high FAF1 samples (Supplementary Fig. 8a). As a control, we confirmed a significant and positive correlation between TβRII and its downstream target P-SMAD2 (Pearson $\chi^2 = 27.44$, $P = 3.37E - 06$; Supplementary Fig. 8c,e); more importantly, activation of AKT (P-AKT 473) positively correlated with TβRII (Fig. 8f; Supplementary Fig. 8c), confirming our finding that AKT activity supports TβRII function and collaborates with TGF-β-induced pro-invasive and pro-metastatic responses in advanced tumours. Our data also highlight the frequent co-activation of AKT and SMAD in breast cancer (Supplementary Fig. 8d,f), suggesting that AKT could cooperate with SMAD to drive malignancy.

## Discussion

TGF-β signalling is hyperactivated in advanced cancers[36]. We found that TβRII protein levels are increased in metastasis, and that TβRII levels determine metastasis severity, raising a novel mechanism by which TGF-β signalling is activated. As the first signalling molecule engaged by the TGF-β ligand, cell surface TβRII is thought to be required for all TGF-β signalling responses. As a low dose of TβRII is sufficient for cytostatic SMAD activation, an increase in TβRII levels contributes to cancer and drug resistance[37], and TβRII expression in CAFs supports cancer growth and survival[38]. Here we provide important new insights into how cell surface TβRII is dynamically regulated in stability and subcellular localization through FAF1-mediated polyubiquitination. We generated FAF1-knockout mice and found that cell membrane TβRII accumulates in FAF1-deficient cells from mouse embryos, confirming that FAF1 restrains TβRII function under physiological conditions. Pathologically, we identified FAF1 as a strong suppressor of TGF-β signalling and showed that it recruits the VCP/βTRCP complex to promote turnover of cell surface TβRII, thereby inhibiting tumorigenic TGF-β function in EMT, invasion and metastasis. This was confirmed by multiple in vivo models of metastasis, an MMTV-PyMT transgenic model of mammary tumour progression and clinical breast cancer samples.

FAF1 is highly expressed in normal epithelial cells or pre-malignant cells (such as MCF7 or MCF10A cells) but is attenuated in the well-established mesenchymal-like MDA-MB-231 cells; thus, it limits the amount of cell surface TβRII to prevent excessive TGF-β responses. Our study revealed that activated AKT and TGF-β cooperate to antagonize the inhibitory effect of FAF1 on TGF-β signalling. When oncogenic AKT is aberrantly activated in cells (for example, by oncogenic mutations or excessive growth factors[39]), AKT can directly phosphorylate FAF1 at Ser 582, which dissociates FAF1 from the cell surface and disrupts its ability to complex with VCP/βTRCP, which is required for the polyubiquitylation of TβRII. This process apparently allows for the activation of both SMAD and non-SMAD pathways. In this context, the tumour-suppressive effects of the SMAD pathway are abrogated by AKT[4]; thus, SMAD-dependent gene responses are advantageous for cancer migration and invasion. The subverted TGF-β/SMAD signalling pathway then actively drives tumour cell progression. Tumour cells with this signature fail to execute TGF-β/SMAD-mediated growth arrest; rather, they undergo EMT in which cells lose cell polarity and cell–cell contacts, become more motile, and acquire fibroblast-like properties. AKT has been shown to reduce SMAD3 function in Hep3B cells in which TGF-β induces an apoptotic response[7,8]. However, SMAD3 levels are often reduced in advanced human tumours, and low SMAD3 levels are sufficient for tumour promotion[40,41]. Aberrant AKT over-activation may therefore redirect TGF-β intracellular signalling, thereby contributing to its switch from tumour suppressor to tumour promoter. Thus, AKT-mediated inactivation of FAF1 protein confirms the high level of cell surface TβRII, which in turn further strengthens both SMAD and AKT (one of the non-SMAD pathways) signals. Thus, a self-enforcing loop directing the tumour-promotive TGF-β pathway is triggered by AKT through the inactivation of FAF1, via which cancer cells are stimulated to invade and metastasize.

FAF1 gene loss was observed to be significantly and inversely correlated with TGF-β3 gene expression. In light of the FAF1 downregulation and the loss of epithelial differentiation that we clearly observed in different breast cancer models, we found the clinical relevance of FAF1 in the percentage of distant metastasis-free survival of breast cancer patients where the cutoff by FAF1 expression correlates with prognosis (Fig. 1j). EMT is only clearly detectable in low frequent breast carcinosarcomas such as the molecular subtype of claudin-low breast cancers. We found that the claudin-low clinical samples have reduced expression of FAF1, further supporting our analysis. Importantly, FAF1 gene loss has been reported in many types of human cancers, suggesting that the tumour-suppressive role of FAF1 is not limited to breast cancer. Moreover, both TβRII and P-SMAD2 levels correlated inversely with the FAF1 level in our breast cancer tissue microarray analysis, further corroborating that FAF1 plays an essential role in controlling TβRII and TGF-β signals. In vivo cancer models using cell lines or transgenic mice consistently showed that loss of FAF1 expression is closely associated with breast cancer metastasis. Either loss of FAF1 or disassembly of the

**Figure 7 | AKT-mediated phosphorylation of FAF1 at Ser 582 functionally disrupts the FAF1-VCP/E3 complex and dismisses FAF1 from targeting cell surface TβRII.** (a,b) Immunoblot (IB) of total cell lysate (TCL) and immunoprecipitants derived from HEK293T cells transfected with Flag-FAF1 and Myr-HA-AKT (a) or Flag-FAF1 wt/S582A/S582D expression plasmids (b) as indicated. *Indicates a non-specific band. (c) IB of TCL and immunoprecipitants derived from Myc-Ub stably expressed HEK293T cells transfected with TβRII-HA and Flag-FAF1 wt/SA/SD expression plasmids as indicated. (d) IB of TCL and biotinylated cell surface TβRII in MDA-MB-231 cells infected with Flag-FAF1 wt/S582A/S582D as indicated. (e) IB of TCL and immunoprecipitants derived from Myc-Ub stably expressed HEK293T cells transfected with TβRII and shFAF1 plasmids and treated with or without LY 294002 (LY, 50 μM for 6 h) and TGF-β (5 ng ml$^{-1}$) at the indicated time points. (f) IB of total TβRII in bone metastatic MDA-MB-231 cells treated with dimethylsulphoxide (DMSO) or N2, N4-dibenzylquinazoline-2,4-diamine (DBeQ; 10 μM) and LY294002 (50 μM) at the indicated time points. (g) IB of TCL and biotinylated cell surface TβRII in bone metastatic MDA-MB-231 cells infected with control (Co.sh) or shFAF1 lentivirus and treated with LY294002 (50 μM) at the indicated time points. (h) MDA-MB-231 cells expressing control vector (Co.vector) or DOX-inducible FAF1 wt/FAF S582A were injected intracardially into mice. DOX was administered 28 days after the inoculation of the cells (as indicated). Metastasis was measured by BLI. Normalized photon flux for the indicated time is presented; error bars, mean ± s.e. P values; Student's t-test (left panel). Representative images (right panel). (i) In our working model, AKT phosphorylates FAF1 to disassociate the membrane FAF1–VCP/E3 complex, thereby stabilizing cell surface TβRII for oncogenic invasion or metastasis (in which AKT activation has already blunted cytostatic TGF-β/SMAD to support oncogenic growth).

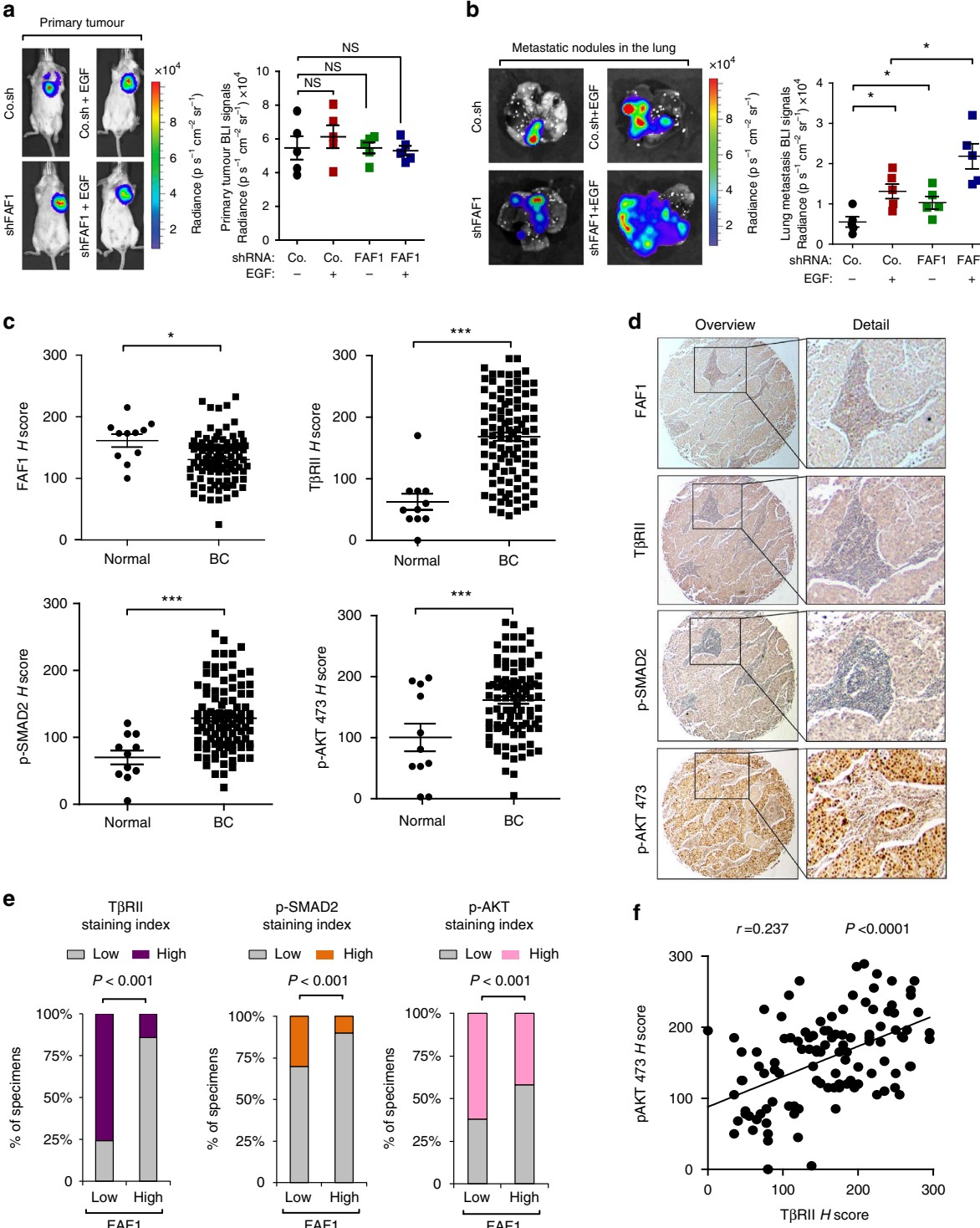

**Figure 8 | Loss of FAF1 correlates with increased TβRII as well as downstream p-SMAD2 and p-AKT expression in patients.** (**a**) Bioluminescent imaging (BLI) of representative mice from each group at week 3 following injection with empty vector (PLKO.1) or 4T1 breast cancer cells stably depleted of FAF1 and pre-treated with or without EGF (5 ng ml$^{-1}$) for 8 h. Ventral images of the mice are shown (left panel). The BLI signal from primary tumours obtained from every mouse in each experimental group is shown. NS, not significant (right panel). (**b**) BLI of representative metastatic nodules in the lung from each experimental group is shown (left panel). Lung metastasis BLI signal for every mouse in each experimental group was analysed and is shown in the right panel (right panel). Data are presented as the mean ± s.d. (**c**–**f**) Immunohistochemistry analysis of FAF1, TβRII, p-SMAD2 and P-AKT 473 in breast cancer tissue microarrays; expression of FAF1 and TβRII, P-SMAD2 and P-AKT 473 in normal ($n = 11$) and breast cancer ($n = 110$) tissues. The data are the mean ± s.e.m. $P$ values obtained using Student's $t$-tests are indicated (**c**). Representative images of each antibody staining are shown. Left panel: objective, × 5; right panel: objective, × 25 (**d**). Percentage of specimens displaying low or high FAF1 expression compared with the expression levels of TβRII, P-SMAD2 and P-AKT (**e**). Scatterplot showing the positive correlation between TβRII and P-AKT 473 expression in patients. Pearson's coefficient tests were performed to assess significance (**f**).

FAF1/VCP/βTRCP complex via AKT phosphorylation can lead to increased breast cancer metastatic potential. We evaluated in the breast cancer tissue microarray specimens and found the status of AKT activation (p-AKT 473 staining) and FAF1 expression (FAF1 staining) were not correlated, suggesting that these two events might be independent of each other.

In metastatic breast cancer cells, blocking PI3K/AKT activity reverses TβRII expression to a very low level (Fig. 7f,g; Supplementary Fig. 7d), demonstrating that TβRII-mediated oncogenic TGF-β signalling is supported by AKT activity. Therapeutic targeting of PI3K/AKT holds significant promise as a treatment for cancer patients[42]. Our results suggest that targeted therapy of the PI3K/AKT pathway may already, in part, target TGF-β signalling. Moreover, combined therapeutics against PI3K/AKT and TβRII might serve as an efficient method for advanced breast cancer patients in the future[4].

In human patients, 70% of breast cancers are oestrogen receptor α (ER) positive and are thus treated with endocrine therapies. However, 25% of these cancers eventually develop endocrine resistance due to *ESR1* gain-of-function mutations, with ligand-independent ER activity driving metastasis[43–45]. In an analysis of >500 breast cancer patients, *ESR1* apparently correlated with the loss of *FAF1*. It is therefore likely that *ESR1* correlates with a gain of TβRII function. Thus, endocrine resistance by *ESR1* could be the result of oncogenic TGF-β activation. Although this hypothesis awaits further investigation, it raises the possibility that combined treatment with a TGF-β antagonist could counteract the development of tolerance to endocrine therapy in ER-positive breast cancer patients.

In closing, our study proposes models of TβRII control and TGF-β functional switch triggered by activation of AKT. It will be interesting to see in the future whether the mechanisms we have unravelled will extend to other cancers. If so, these hitherto underappreciated roles of TGF-β in malignancy and drug resistance could serve as a foundation for improving therapeutics against devastating cancers.

## Methods
**Animal studies.** All mouse experiments were approved by and performed following the guidelines of the Institutional Animal Care and Use Committee at Zhejiang University.

**Nude mice and cell culture.** Nude mice were purchased from the animal husbandry centre of the Shanghai Institute of Cell Biology, Academia Sinica, Shanghai, China. HEK293T cells, HeLa cervical cancer cells, MCF7, MDA-MB-231, MDA-MB-435 breast cancer cells and A549 lung cancer cells from Leiden University Medical Center[46,47] were cultured in DMEM supplemented with 10% fetal bovine serum and 100 U ml$^{-1}$ penicillin–streptomycin. MCF10A (M1), MCF10A-RAS (M2), M3 and M4 cells were cultured in DMEM/F12 supplemented with 5% horse serum, 20 ng ml$^{-1}$ EGF, 0.5 μg ml$^{-1}$ hydrocortisone, 100 ng ml$^{-1}$ cholera toxin and 100 U ml$^{-1}$ penicillin–streptomycin.

**Generation of Faf1$^{-/-}$ mice.** Faf1$^{flox/+}$ mice (C57BL/6J) were generated by standard homologous recombination at Shanghai Biomodel Organism, Shanghai, China. In these mice, *Faf1* exon 3 was flanked by *loxP* sequences. Faf1$^{flox/+}$ mice were then mated to *EIIa-Cre* transgenic mice (FVB/N; Jackson Laboratory), in which the adenovirus *EIIa* promoter directs the expression of Cre enzyme in early mouse embryos (two- to eight-cell stage) to achieve homologous recombination between *LoxP* sites, thereby triggering the deletion of exon 3 in all cells of the developing animal, including the germ cells that transmit the genetic alteration to progeny. Deletion of exon 3 results in loss of the signal peptide and disrupts its ORF, leading to the loss of *Faf1* expression. The first generation of *EIIa-Cre*; Faf1$^{flox/+}$ mice might be chimeric due to the mosaic activity of Cre recombinase. Therefore, chimeric offspring were backcrossed with C57BL/6J to generate Faf1$^{+/-}$ mice, which were then intercrossed for the production of *Faf1*-deficient (Faf1$^{-/-}$) mice. Mouse genotyping was performed using genomic DNA isolated from mouse tails by PCR with the following primers: 5′-CAGCCCACAACT CACCTTTT-3′ (faf1-KO-P1); 5′-AATTGAAGGCCAGACGTAGC-3′ (faf1-KO-P3-W686); 5′-CTGAGCCCAGAAAGCGAAGGA-3′ (Neo-R). A 1,086-bp and ∼686-bp fragment were produced for the WT and null alleles, respectively.

**Mouse metastasis assay.** MDA-MB-231-Luc and A549-Luc cells[32] were used to stably overexpress or were depleted of target genes by lentivirus infection and puromycin selection for 3 days. A single-cell suspension of these cells (1 × 10$^5$/100 μl of PBS) was inoculated into the left heart ventricle ($n > 5$) according to the method described by Arguello et al.[48]. The development of metastases was monitored weekly by bioluminescent reporter imaging. After 6 weeks (or according to the description in the figure legend), the mice were killed, and the metastasis nodules were dissected. For the induced model, doxycycline was administered to mice at indicated time point after injection of the cancer cells, through the diet (625 mg kg$^{-1}$ of food) as well as by intraperitoneal injection (25 mg kg$^{-1}$ of body weight; three times a week). For the MMTV-PyMT models, FAF1 KO mice were backcrossed to FVB background for more than six generations before breeding with MMTV-PyMT transgenic mice (Jackson Laboratory) in FVB background.

For spontaneous tumorigenesis and metastasis studies, female mice carrying the specific oncogenes were examined weekly for mammary tumours. Tumours were considered established when they became palpable for 2 consecutive weeks, and tumours were measured by calipers for calculation of tumour volumes ($\pi \times length \times width^2/6$). Lung nodules were counted directly after fixation or after sectioning and staining of the lungs. For orthotopic primary tumour formation, female FVB (for PyMT tumour cells experiments) mice at 6 weeks old were anaesthetized and a small incision was made to reveal the mammary gland. MMTV-PyMT tumour cells (2.5 × 10$^4$) resuspended in 10 μl PBS were injected directly into the mammary fat pad. The primary tumour growth was monitored weekly by measurement of the tumour size. Experimental metastasis to the lungs was induced by injecting 0.2 × 10$^6$ cells in 100 μl of PBS in the tail vein of female athymic nude mice. Mouse nipple implantation of 4T1 cells was based on a previously published method[49]. Female BALB/c mice were anaesthetized and used for this assay. A total of 1 × 10$^5$ 4T1-Luc cells were injected through the nipple area into the mammary fat pad. At 21 days after injection, luciferin was injected and the primary tumours were analysed, then the mice were killed and analysed for acquisition of secondary tumour(s). All primary/metastatic tumours were detected by bioluminescent imaging with the IVIS 100 (Caliper Life Sciences, Hopkinton, MA, USA). The bioluminescent imaging signal intensity was quantified as the sum of photons within a region of interest given as the total flux (photons per second).

**GSEA.** We used GSEA v2.0 to perform GSEA on various functional and/or characteristic gene signatures. Gene sets were obtained from the MSigDB database v3.0 (September 2010 release). Statistical significance was assessed by comparing the enrichment score to enrichment results generated from 1,000 random permutations of the gene set to obtain P values (nominal P value). Data from NKI 295, a well-annotated human breast cancer database, were analysed for the enriched gene signature, and FAF1-high (FAF1 ≥ 0.28, $n = 103$) versus FAF1-low (FAF1 ≤ − 0.34, $n = 104$) samples were compared.

**Plasmids and reagents.** FAF1 and related expression constructs were cloned and verified by DNA sequencing. FAF1 wt and FAF1 UBX domain-deleted mutant fragments were subcloned into the pLv-bc-puro lentivirus construct. FAF1 S582A and other mutants were generated by site-directed mutagenesis and confirmed by DNA sequencing. Myr-AKT1 constructs were kindly provided by P. Coffer (University Medical Center Utrecht, The Netherlands). The reagents used were IGF-1 (R&D 291-G1), LY294002 (Cell Signalling), MG132 (Selleck, catalog no. S2619), CHX (Sigma, C104450), SB431542 (Millipore, 616461) and λ-phosphatase (Biolabs). BKM120, MK2206 and GDC0068 were purchased from Selleck. The recombinant proteins used were human active AKT1 protein (R&D, 1775-KS), TβRII-ICD (Sino Biological Inc., 10358) and TGF-β (Sino Biological Inc., 10804).

**Immunoprecipitation and immunoblotting.** Cells were lysed with 1 ml of lysis buffer (20 mM Tris-HCl pH 7.4, 2 mM EDTA, 25 mM NaF and 1% Triton X-100) containing protease inhibitors (Sigma) for 10 min at 4 °C. After centrifugation at 12 × 10$^3$g for 15 min, the protein concentrations were measured, and equal amounts of lysate were used for immunoprecipitation. Immunoprecipitation was performed with different antibodies and protein A-Sepharose (GE Healthcare Bio-Sciences AB) for 3 h at 4 °C. Thereafter, the precipitants were washed three times with washing buffer (50 mM Tris-HCl pH 8.0, 150 mM NaCl, 1% Nonidet P-40, 0.5% sodium deoxycholate and 0.1% SDS), and the immune complexes were eluted with sample buffer containing 1% SDS for 5 min at 95 °C. The immuno-precipitated proteins were then separated by SDS–polyacrylamide gel electrophoresis (SDS–PAGE). Western blotting was performed with specific antibodies and secondary anti-mouse or anti-rabbit antibodies conjugated to horseradish peroxidase (Amersham Biosciences). Visualization was achieved with chemiluminescence. For proteins that migrated close to the IgG heavy chain, protein A-horseradish peroxidase was used. Biotinylation analysis of cell surface receptors was performed as previously described[46,47], the details are shown below. The cells were biotinylated for 40 min at 4 °C and then incubated at 37 °C for the indicated times. The biotinylated cell surface receptors were precipitated with streptavidin beads and analysed by immunoblotting. The antibodies used for immunoprecipitation, immunoblotting and immunofluorescence were as follows: AKT at 1:5,000 (IB) and 1:250 (IP; no. 2938, Cell Signalling); phospho-AKT substrate (RXRXXS*-T*) at 1:1,000 (IB; no. 10001, Cell Signalling); phospho-AKT (Ser 473) at 1:1,000

(IB; no. 9271, Cell Signalling); TβRII at 1:1,000 (IB) and 1:50 (IP; L-21, Santa Cruz); N-cadherin at 1:50,000 (IB; 610920, BD); tubulin at 1:1,000 (IB; no. 2146, Cell Signalling); SMAD4 at 1:1,000 (IB; B8, Santa Cruz); SMAD2-3 at 1:2,500 (IB) and 1:500 (IP; 610842 BD); phospho-SMAD2 at 1:5,000 (IB; no. 3101, Cell Signalling); p38 at 1:3,000 (IB; no. 535, Santa Cruz); p-p38 at 1:2,000 (IB; no. 4511, Cell Signalling); Ub at 1:1,000 (IB; P4D1 Santa Cruz); fibronectin at 1:1,000 (IB; Sigma); SMA (Sigma); vimentin at 1:1,000 (IB; no. 5741 Cell Signalling); E-cadherin at 1:10,000 (IB; BD 610181); β-actin at 1:10,000 (IB; A5441, Sigma); c-Myc at 1:1,000 (IB; a-14, sc-789, Santa Cruz Biotechnology); HA at 1:1,000 (IB; Y-11, sc-805, Santa Cruz Biotechnology); HA at 1:10,000 (IB; 12CA5, home-made); Flag at 1:10,000 (IB; M2, Sigma); FAF1 (Bethyl Laboratories); VCP at 1:2,000 (IB; no. 2648, Cell Signalling); Ufd1 at 1:1,000 (IB; no. 13789, Cell Signalling); Npl4 (IB; no. 13489, Cell Signalling); Cul1 at 1:1,000 (IB; no.4995 Cell Signalling); and βTRCP at 1:1,000 (IB; no. 4394, Cell Signalling). All the uncropped scans of the western blots are shown in Supplementary Fig. 9.

**Lentiviral transduction and the generation of stable cell lines.** Lentiviruses were produced by transfecting HEK293T cells with shRNA-targeting plasmids and the helper plasmids pCMV-VSVG, pMDLg-RRE (gag/pol) and pRSV-REV. The cell supernatants were collected 48 h after transfection and were either used to infect cells or stored at −80 °C. To obtain stable cell lines, cells were infected at low confluence (20%) for 24 h with lentiviral supernatants diluted 1:1 in normal culture medium in the presence of 5 ng ml$^{-1}$ of polybrene (Sigma). At 48 h after infection, the cells were placed under puromycin selection for 1 week and then passaged before use. Puromycin was used at 1 μg ml$^{-1}$ to maintain MDA-MB-231, MCF10A and HaCaT cells. Lentiviral shRNAs were obtained from Sigma (MISSION shRNA). Typically, five shRNAs were identified and tested, and the two most effective shRNAs were used for the experiment. We used the following shRNAs:

TRCN000000424 (1#) and TRCN000000424 (#2) for human FAF1 knockdown; TRCN0000004 250 (1#) and TRCN0000004 252 (#2) for human VCP knockdown; TRCN0000006541 + TRCN0000315200 mixture for Human βTRCP knockdown; TRCN000039793 + TRCN000039797 + TRCN000010162 + TRCN0000010174 mixture for Human AKT1 knockdown; and TRCN0000191433 for mouse FAF1 knockdown.

**Transcription reporter assay.** HEK293T cells were seeded in 24-well plates and transfected with the indicated plasmids using calcium phosphate. At 24 h after transfection, the cells were treated with TGF-β overnight or left untreated and then collected. Luciferase activity was measured with a PerkinElmer luminometer. The internal transfection control Renilla expression plasmid (10 ng) was used to normalize luciferase activity. Each experiment was performed in triplicate, and the data represent the mean ± s.d. of three independent experiments.

**Quantitative real-time PCR.** Total RNA samples were prepared using a NucleoSpin RNA II kit (Biospin). A total of 1 μg of RNA was reverse-transcribed using the RevertAid First Strand cDNA Synthesis kit (Fermentas). Real-time PCR was conducted with SYBR Green (Applied Bioscience) using a StepOne Plus real-time PCR system (Applied Bioscience). Target gene expression values were normalized to 18S RNA levels. All primers used in quantitative PCR with reverse transcription are listed in Supplementary Table 3.

**Ubiquitination assay.** Cells were washed with PBS and lysed in two pellet volumes of RIPA buffer (20 mM NAP, pH7.4, 150 mM NaCl, 1% Triton, 0.5% sodium-deoxycholate and 1% SDS) supplemented with protease inhibitors and 10 mM N-ethylmaleimide. The lysates were sonicated, boiled at 95 °C for 5 min, diluted in RIPA buffer containing 0.1% SDS and centrifuged at 4 °C (16 × 10$^3$g for 15 min). The supernatant was incubated with specific antibodies and protein A-Sepharose for 3 h at 4 °C. After extensive washing, bound proteins were eluted with 2 × SDS sample buffer and separated on SDS–PAGE followed by western blotting[47]. For the detection of TβRII ubiquitination, cells were treated with the lysosome inhibitor bafilomycin A1 (1 μM) for 6 h before they were collected for the ubiquitination assay.

**Cellular fractions.** Cytosolic, membrane and nuclear fractions were prepared using the ProteoExtract kit (Calbiochem) according to the manufacturer's instructions.

**Pulse chase.** As previously described[46], cells were plated in six-well plates. Cells were starved for 3 h in Met/Cys-free medium and pulsed for 40 min with 200 μCi ml$^{-1}$ of $^{35}$SMet/Cys. After two washes, the cells were chased in medium supplemented with cold (unlabelled) Met and Cys (100 μg ml$^{-1}$) before collecting. Endogenous proteins were collected from the extracts by immunoprecipitation, then were resolved by SDS–PAGE and visualized by autoradiography. Each experiment was performed in duplicate.

**Mass spectrometry.** SDS–PAGE gels were minimally stained with Coomassie brilliant blue, cut into six molecular weight ranges based on heavy-chain IgG bands, and digested with trypsin. Immunocomplexes were identified on a Thermo Fisher LTQ (majority) or Velos-Orbitrap mass spectrometer. Spectral data were then searched against the human protein RefSeq database in BioWorks or the Proteome Discoverer Suites using either SeQuest (for LTQ data) or Mascot (Orbitrap data) software. The immunoprecipitation–mass spectrometry results were transferred into a FileMaker-based relational database generated in-house, where protein identification numbers (protein GIs) were converted to GeneID identifiers according to the NCBI 'gene accession' table[31].

**Immunofluorescence.** For the EMT assay, HaCaT cells were plated onto collagen I-coated coverslips. After adhesion, the cells were serum starved overnight and either left untreated or treated 36 h with TGF-β (2 ng ml$^{-1}$) in 2% fetal bovine serum. Cells were fixed for 10 min in 4% paraformaldehyde in PBS, permeabilized with 0.2% TritonX100-PBS and then blocked with 3% bovine serum albumin (BSA) in PBS for 30 min at room temperature. Anti-E-cadherin (BD Transduction Laboratories) was used at a dilution of 1:1,000 in 3% BSA-PBS and incubated for 3 h at room temperature. Secondary, AlexaFluor488-labelled anti-mouse antibody (Molecular Probes) was used at a dilution of 1:200 in 3% BSA-PBS, together with Phalloidin (Molecular Probes) and incubated for 1 h at room temperature. Coverslips were mounted with 4′,6-diamidino-2-phenylindole-containing VECTASHIELD mounting medium (Vector Laboratories Inc.). Fluorescence was recorded using a Zeiss Axioplan microscope.

For localization, HeLa cells were transfected with target plasmids and then processed as described above with a specific primary antibody and secondary AlexaFluor488-labelled and AlexaFluor593-labelled anti-rabbit antibodies (Molecular Probes).

**Three-dimensional spheroid invasion assays.** Semi-confluent MCF10A-RAS (MII) cells were trypsinized, counted and re-suspended in medium containing 2.4 mg ml$^{-1}$ methylcellulose (Sigma) at a concentration of 10$^4$ cells per ml. A total of 100 μl of suspension was added into each well of a U-bottom 96-well-plate, allowing for the formation of one spheroid per well. All spheroids consisted of 10$^3$ cells. Two days after plating, the spheroids were collected and embedded into collagen. A flat-bottom 96-well-plate was coated with neutralized bovine collagen-I (PureCol, Advanced BioMatrix) according to the manufacturer's protocol. Single spheroids were embedded in a 1:1 mix of neutralized collagen and medium supplemented with 12 mg ml$^{-1}$ of methylcellulose. TGF-β (2 ng l$^{-1}$) was directly added to the embedding solution. Invasion was monitored over the subsequent 2 days and quantified by measuring the area occupied by the cells using ImageJ software. Images were captured at days 0, 1 and 1.5 after embedding.

**Migration assays.** Transwell assays were performed in 24-well PET inserts (Falcon 8.0-μm pore size) for migration assays. MDA-MB-231 cells with targeted gene manipulations were serum starved overnight. Then, 5 × 10$^4$ or 10 × 10$^4$ cells were plated in transwell inserts (at least three replicas for each sample) and left treated with or without TGF-β (5 ng ml$^{-1}$) for 8 h. Cells in the upper part of the transwells were removed with a cotton swab; migrated cells were fixed in 4% paraformaldehyde and stained with 0.5% crystal violet. The filters were imaged, and the total number of cells was counted. Every experiment was repeated at least three times independently.

**Purification of bacterially expressed recombinant FAF1 proteins.** GST-FAF1 wt and mutant expression constructs were generated by sub-cloning into pGEX-4T1 vectors. Plasmids were used to transform the Escherichia coli strains BL21 and Rosetta, respectively. Cultures were grown overnight at 37 °C. The next day, the cultures were diluted 1:50 in fresh Luria Bertani medium and grown at 37 °C to an OD600 of 0.6. The cells were then induced overnight at 24 °C in the presence of 0.5 mM isopropyl-β-D-thiogalactopyranoside, 20 mM HEPES pH 7.5, 1 mM MgCl$_2$ and 0.05% glucose. For purification, the washed pellets were resuspended in lysis buffer (PBS, 0.5 M NaCl, complete protease inhibitors (Roche), 1 mM phenylmethanesulfonyl fluoride, 1% Triton X-100). After sonication and a freeze–thaw step, the supernatants of the cell lysates were incubated with glutathione Sepharose beads (GE Healthcare). The beads were washed twice with lysis buffer and three times with 50 mM Tris-HCl pH 7.5, 0.5 M NaCl. Purified proteins were eluted in 50 mM Tris-HCl pH 7.5, 0.5 M NaCl and 20 mM glutathione. For protein purification, GST was removed using biotin-tagged thrombin.

**Immunohistochemical staining and evaluation.** Formalin-fixed paraffin-embedded microarrays of breast cancer tissues were obtained from US Biomax (BC081120). Primary antibodies specific to p-SMAD2 (1:50; Cell Signalling 3108), FAF1 (1:200; Bethyl), TβRII (1:100; Santa Cruz) and p-Akt Ser 473 (1:200; Cell Signalling #9271) were used for immunohistochemical staining. The quantification of staining was expressed as an H score. The H score was determined by the formula 3 × the percentage of strongly staining cells + 2 × the percentage of moderately staining cells + the percentage of weakly staining cells, yielding a range of 0 to 300.

**Statistical analyses.** Statistical analyses were performed with a two-tailed unpaired $t$-test or as indicated in the legends. $P$ value is indicated by asterisks in the Figures: $*P < 0.05$; $**P < 0.01$; $***P < 0.001$. Differences at $P = 0.05$ and lower were considered significant.

**Genotyping of $Faf1^{-/-}$ mice, cDNA screen and migration/invasion assays.** Associated references are available in the Supplementary Information.

**Data availability.** The authors declare that all data supporting the findings of this study are available within the article and its Supplementary Information files.

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

## Acknowledgements

We are grateful to Maarten van Dinther and Midory Thorikay for technical assistance. We are grateful to Martijn Rabelink for shRNA constructs. We thank Kohei Miyazono for reagents. This work was supported by special program from Ministry of Science and Technology of China (2016YFA0502500), the National Natural Science Foundation of China (grant numbers 31471315, 31671457, 31571460, R14C070002, K124924615, 81470851, 81572651 and LR15C060001), Jiangsu Natural Science Foundation (BK20150354), the Fundamental Research Funds for the Central Universities (2016QN81013) and the Cancer Genomics Centre Netherlands.

## Author contributions

F.Z., F.X. and K.J. designed and performed the experiments, analysed and interpreted the results, and wrote the manuscript. L.S. performed the analysis of breast cancer patient data for GSEA. Y.L. and S.W. assisted in animal experiments. B.Y. contributed to the mass spectroscopy analysis. H.v.D, J.L.J. and J.J. were involved in data analysis. L.Z., Y.F., Y.T. directed the research, P.t.D., H.W., S.D. interpreted the data and wrote the manuscript.
