## [Peer Review File · Nature Communications]

Reviewers' comments:

Reviewer #1 (Remarks to the Author):

The new paper by Zhou et al comes from a leading laboratory in the fields of TGF-beta signal transduction, EMT, angiogenesis and cancer metastasis. Transforming growth factor beta (TGF-beta) signal transduction initiates when this extracellular growth factor associates with its cell surface receptor known as TbetaRII (type II receptor). This initiates a signaling cascade that sequentially activates a second receptor (type I), Smad proteins and many other adaptor proteins and protein or lipid kinases. Much is known about this signaling cascade, yet the regulation of the initial key component, the TbetaRII, is the least studied aspect in this family of signaling proteins. Zhou et al explain how the cell surface levels of TbetaRII are regulated. They identify the adaptor protein FAF1 as a key regulator of this process. And then they provide a rather complete picture of the mechanism and also series of in vitro and in vivo biological experiments that convincingly show that the relative presence of FAF1 controls the presence of TbetaRII and the consequent oncogenic potential of various tumor cells. What is particularly commendable in this paper is that the mechanistic work went all the way into identifying that the Akt kinase phosphorylates FAF1, and thus promotes its dissociation from the TbetaRII, causing enhanced TGF-beta signaling. Downstream of TGF-beta receptor signaling acts the Akt kinase, which then becomes super-activated and continues, in a sustainable manner to promote TGF-beta signaling via the stabilized TbetaRII receptor. This mechanism explains long-standing questions in the field of TGF-beta in cancer, and provides one of the best so far known mechanisms that make TGF-beta an oncogenic signaling pathway in tumors.

The work is extensive, very well performed, with massive number of data and mouse models, xenografts, allografts and genetic models of cancer including the knockout mouse for FAF1. These models are nicely coupled to the analysis of human cancers, providing the necessary correlative evidence for the relevance of this mechanism in human cancer. The only comments I can offer to this otherwise excellent paper is a number of small typographical errors and small necessary changes in the figures, which are essential to be corrected, and the lack of certain methodological details that should be added prior to publications.

Methods: not all mouse tumor/metastasis assays are presented in the methods. The PyMT mouse I could not find in the methods, neither its breeding to the FAF1 knockout. The EGF tumor model may also be missing, please check. All in vivo experiments should be presented in the methods.

Minor but essential mistakes in the main figures:

Figure 1e: what happens in lane 5 of this western blot? Why is the TbetaRII level enhanced in the absence (-) of NH4Cl? The corresponding input lysate in lane 2 of the same blot shows almost undetectable TbetaRII protein. This should be explained or corrected.

Figure 2c: the time units are in min not hrs.

Figure 2d: the label: long exposure is also needed next to the second P-SMAD2 blot. The graph that quantifies the western blot should show time units in hrs not min, correct?

Figure 3g and 7d: label on top: Biontin-labelled should read: biotin-labelled.

Figure 7i: it would be very useful if the cartoon indicated that non-SMAD incorporates AKT, leading to the positive amplification loop of AKT-TGF-beta signaling.

Figure 8d: the "representative" IHC figure is not explained in the legend. We assume that this is BC and not normal, showing low FAF1, high TbetaRII, high P-SMAD2 and high p-AKT. This is not obvious to the reader. The brown staining is not clearly discernible without comparison to the normal breast sample. This figure panel requires some further detail to help the reader understand what is seen. It would be ideal to show a figure that reflects the data of figure 8e, comparing low and high FAF1 staining.

Reviewer #2 (Remarks to the Author):

Zhou et al demonstrated that Fas-associated factor 1 (FAF1) controls EMT and metastasis in breast cancer cells by regulating the membrane levels of TGF- β type II receptor (T β RII) via a VCP/ β TRCP complex-dependent ubiquitylation process. The authors initially demonstrated that TGF- β signaling through increased T β RII membrane accumulation lead to breast cancer metastasis. They further demonstrated that T β RII membrane levels were controlled by lysosomal, not proteolytic, degradation. Using immunoprecipitation/mass spectrometry analysis in the presence of lysosomal inhibitors, FAF1 was found to co-IP with T β RII. The authors further demonstrate that FAF1 is downregulated in metastatic breast cancers and its levels correlate with prognosis. Downregulation of FAF1 in breast cancer cells caused an increase accumulation of T β RII at the membrane causing EMT and metastasis. Overexpression of FAF1 led to decreased T β RII membrane accumulation and inhibited metastasis. FAF1 was shown to regulate T β RII lysosomal degradation via its UBX domain, which interacted with the VCP/ β TRCP complex and caused ubiquitylation of T β RII. Ubiquitylated T β RII was internalized from the membrane and degraded by the lysosome. Using xenograft, orthograft and GEM models of breast cancer, the authors demonstrated that loss of FAF1 increased cancer metastases through increased TGF β signaling. Finally, the authors demonstrated that AKT kinase could phosphorylate FAF1 directly, causing the FAF1-VCP/ β TRCP complex to disassemble, allowing for T β RII membrane accumulation and breast cancer metastasis. Together these data demonstrate that loss of FAF1 or AKT phosphorylation of FAF1 leads to increased TGF- β signaling and breast cancer metastasis through T β RII membrane accumulation.

Concerns

1. The authors do not address how FAF1 levels are decreased during breast cancer progression? Is it lost through a genomic event or is the FAF1 locus epigenetically regulated?
2. While the investigators have demonstrated that EGF can stimulate AKT activity in breast cancer cells, thereby leading to disassembly of the FAF1/VCP/ β TRCP complex, T β RII membrane accumulation and metastasis, they should also address whether estrogen and progesterone, relevant mitogens that also affect breast cancer cell properties, also lead to AKT activation and FAF1 inhibition.
3. Loss of either FAF1 or disassembly of the FAF1/VCP/ β TRCP complex via AKT phosphorylation can each lead to increased breast cancer metastatic potential. Since loss of FAF1 or AKT activation would both lead to increased T β RII levels and TGF- β signaling/metastasis, these events should be redundant and therefore potentially mutually exclusive. The authors should evaluate this possibility in their tissue microarray breast cancer specimens.
4. On page 12 and elsewhere, the authors use a variety of designations for the Faf1 knockout mice, i.e., sometimes in all caps and sometimes with all lower case letters, which should be corrected

Reviewer #3 (Remarks to the Author):

The manuscript by Zhou et al. entitled 'FAF1 phosphorylation by AKT results in TGF-beta type II receptor accumulation on the cell surface and drives breast cancer metastasis' describes the identification of a novel adapter protein of the TGF-beta type II receptor and the VCP/E3 ligase complex. As such FAF1 turns out to be an important regulator of cell surface TGF-beta type II receptor controlling the activation level of TGF-beta induced signaling. PI3K/AKT signaling antagonizes TGF-beta-induced growth arrest and high levels of TGF-beta in tumours correlate with strong PI3K/AKT signaling. The authors show elegantly that growth factor-induced AKT activity specifically phosphorylates FAF1 resulting in the dissociation of FAF1 from the plasma membrane and its association with the TGF-beta II receptor reinforcing the pro-metastatic functions induced by TGF-beta in different breast cancer models. The authors show that FAF-1 expression levels associate with metastasis free survival of patients. All together the authors position the FAF1-VCP/E3 complex as a controlling point contributing to the shift between normal TGF-beta cytostatic

activity towards oncogenic invasive growth and metastasis. The manuscript is well written and experiments are expertly obtained. The following suggestions/remarks are given to expedite publication in Nature communications:

- In Fig 5D the authors explain that in MEFs with graded expression of FAF1, the FAF1+/+ MEFS are not expressing EMT inducing transcription factors and are strongly E-cadherin positive. Most MEFs are co-expressing EMT controlling nuclear factors, mesenchymal markers and low E-cadherin expression. The expression bar should be adapted or expression data should be properly normalized and represented as now MEFs are presented as epithelial cells.
- In the discussion the authors mention that Snail is directly regulating the Faf1 promoter which is an important result as it could explain the well-established correlation of EMT inducing transcription factors that facilitate enhanced TGF-beta signaling. These data should be implemented in the manuscript or otherwise the Snail part left out of the discussion. There have been many target genes and differential expressed genes associated with Snail expression published but Faf1 has so far not been reported/listed as a potential candidate target gene. In this context the authors link Snail expression and the general repression of Faf1 in human cancer. This should be tuned down as Snail expression is in most cancer cases hardly expressed and cannot be the reason for the so called broad repression of Faf1.
- Percentage of distant metastasis free survival (Fig 1J) should be explained better in light of the link of FAF1 downregulation and loss of epithelial differentiation which was clearly established by the authors in this study using different breast cancer models. Are the FAF1 low tumors ER-, PR-negative? Furthermore, for human breast cancer besides the potential broad existence of partial EMT, EMT is only clearly detectable in low frequent breast carcinosarcomas and in the molecular subtype of claudin low breast cancers. As the authors link FAF1 with invasion and metastasis through modulation of EMT they should examine if the FAF1 expression level is more attenuated in these well-established EMT like breast cancers.

Reviewers' comments:

Reviewer #1 (Remarks to the Author):

The new paper by Zhou et al comes from a leading laboratory in the fields of TGF-beta signal transduction, EMT, angiogenesis and cancer metastasis. Transforming growth factor beta (TGF-beta) signal transduction initiates when this extracellular growth factor associates with its cell surface receptor known as TbetaRII (type II receptor). This initiates a signaling cascade that sequentially activates a second receptor (type I), Smad proteins and many other adaptor proteins and protein or lipid kinases. Much is known about this signaling cascade, yet the regulation of the initial key component, the TbetaRII, is the least studied aspect in this family of signaling proteins. Zhou et al explain how the cell surface levels of TbetaRII are regulated. They identify the adaptor protein FAF1 as a key regulator of this process. And then they provide a rather complete picture of the mechanism and also series of in vitro and in vivo biological experiments that convincingly show that the relative presence of FAF1 controls the presence of TbetaRII and the consequent oncogenic potential of various tumor cells. What is particularly commendable in this paper is that the mechanistic work went all the way into identifying that the Akt kinase phosphorylates FAF1, and thus promotes its dissociation from the TbetaRII, causing enhanced TGF-beta signaling. Downstream of TGF-beta receptor signaling acts the Akt kinase, which then becomes super-activated and continues, in a sustainable manner to promote TGF-beta signaling via the stabilized TbetaRII receptor. This mechanism explains long-standing questions in the field of TGF-beta in cancer, and provides one of the best so far known mechanisms that make TGF-beta an oncogenic signaling pathway in tumors.

The work is extensive, very well performed, with massive number of data and mouse models, xenografts, allografts and genetic models of cancer including the knockout mouse for FAF1. These models are nicely coupled to the analysis of human cancers, providing the necessary correlative evidence for the relevance of this mechanism in human cancer. The only comments I can offer to this otherwise excellent paper is a number of small typographical errors and small necessary changes in the figures, which are essential to be corrected, and the lack of certain methodological details that should be added prior to publications.

Answer: Many thanks for your comments, we have corrected those mistakes.

Methods: not all mouse tumor/metastasis assays are presented in the methods. The PyMT mouse I could not find in the methods, neither its breeding to the FAF1 knockout. The EGF tumor model may also be missing, please check. All in vivo experiments should be presented in the methods.

Answer: We have added missing information in the method: "Mouse metastasis assay" in Page 23 second paragraph and also in Page 24. Many thanks.

Minor but essential mistakes in the main figures:

Figure 1e: what happens in lane 5 of this western blot? Why is the TbetaRII level enhanced in the absence (-) of NH₄Cl? The corresponding input lysate in lane 2 of the same blot shows almost undetectable TbetaRII protein. This should be explained or corrected.

Response: We used limited amount of TβRII antibody to perform the immunoprecipitation.

The antibody was saturated by the endogenous TβRII protein either in the absence (-) or presence (+) of NH4Cl. Only in this way, we could pull down equal amount of TβRII then compared their binding affinity to endogenous FAF1. This has been explained in the text at Page 5, first paragraph. Many thanks.

Figure 2c: the time units are in min not hrs.

Response: This was corrected.

Figure 2d: the label: long exposure is also needed next to the second P-SMAD2 blot. The graph that quantifies the western blot should show time units in hrs not min, correct?

Response: They were corrected.

Figure 3g and 7d: label on top: Biotin-labelled should read: biotin-labelled.

Response: They were corrected.

Figure 7i: it would be very useful if the cartoon indicated that non-SMAD incorporates AKT, leading to the positive amplification loop of AKT-TGF-beta signaling.

Response: This has been added in the figure and also in the text (Page 17, the end of first paragraph).

Figure 8d: the “representative” IHC figure is not explained in the legend. We assume that this is BC and not normal, showing low FAF1, high TbetaRII, high P-SMAD2 and high p-AKT. This is not obvious to the reader. The brown staining is not clearly discernible without comparison to the normal breast sample. This figure panel requires some further detail to help the reader understand what is seen. It would be ideal to show a figure that reflects the data of figure 8e, comparing low and high FAF1 staining.

Response: We have provided control high FAF1 staining as Supplementary Figure S8a. Related description has been added in the text in Page 18, first paragraph. Many thanks.

Reviewer #2 (Remarks to the Author):

Zhou et al demonstrated that Fas-associated factor 1 (FAF1) controls EMT and metastasis in breast cancer cells by regulating the membrane levels of TGF-β type II receptor (TβRII) via a VCP/βTRCP complex-dependent ubiquitylation process. The authors initially demonstrated that TGF-β signaling through increased TβRII membrane accumulation lead to breast cancer metastasis. They further demonstrated that TβRII membrane levels were controlled by lysosomal, not proteolytic, degradation. Using immunoprecipitation/mass spectrometry analysis in the presence of lysosomal inhibitors, FAF1 was found to co-IP with TβRII. The authors further demonstrate that FAF1 is downregulated in metastatic breast cancers and its levels correlate with prognosis. Downregulation of FAF1 in breast cancer cells caused an increase accumulation of TβRII at the membrane causing EMT and metastasis. Overexpression of FAF1 led to decreased TβRII membrane accumulation and inhibited metastasis. FAF1 was shown to regulate TβRII lysosomal degradation via its UBX domain, which interacted with the VCP/βTRCP complex and caused ubiquitylation of TβRII. Ubiquitylated TβRII was internalized from the membrane and

degraded by the lysosome. Using xenograft, orthograft and GEM models of breast cancer, the authors demonstrated that loss of FAF1 increased cancer metastases through increased TGF β signaling. Finally, the authors demonstrated that AKT kinase could phosphorylate FAF1 directly, causing the FAF1-VCP/ β TRCP complex to disassemble, allowing for T β RII membrane accumulation and breast cancer metastasis. Together these data demonstrate that loss of FAF1 or AKT phosphorylation of FAF1 leads to increased TGF- β signaling and breast cancer metastasis through T β RII membrane accumulation.

Answer: Many thanks for your comments.

Concerns:

1. The authors do not address how FAF1 levels are decreased during breast cancer progression? Is it lost through a genomic event or is the FAF1 locus epigenetically regulated?

Response: So far, we do not have clear answer. As shown below, we found that the FAF1 gene is lost or even deleted in many tumor types as revealed by GISTIC analysis, the total percentage is high up to about 30% in pancreatic cancer, liver cancer and lung cancer. In TCGA breast cancer analysis, 29% of the 963 patients showed loss of FAF1 gene copies. Therefore, it could be a genomic event but the detailed mechanism requires further investigation. We mentioned this in the discussion at Page 21.

Combined with our analysis, these observations strongly favor the notion that the general loss of FAF1 expression could result in cancer progression.

	Patient dataset	n=	Deletion (%)	Loss (%)
1	ChRCC (TCGA)	65	1.5	80
2	PCPG (TCGA)	162	1.9	70.4
3	Pancreas (UTSW)	109	1.8	38.5
4	Lung squ (TCGA)	178		21.3
5	Liver (TCGA)	366		29
6	Glioma (TCGA)	283	2.1	33.9
7	ACC (TCGA)	88	1.1	34.1
8	Breast (TCGA)	963		29
9	Uveal melanoma (TCGA)	80		28.7

10	NSCLC (TCGA)	1144		27.5
11	Esophagus (TCGA)	184		29.9
12	Prostate (FHCRC,2016)	136	1.5	27.9
13	Colorectal (TCGA)	220		25.5
14	Melanoma (TCGA)	287		17.4
15	Stomach (TCGA)	393	0.5	19.6
16	Ovarian (TCGA)	311		17.4

2. While the investigators have demonstrated that EGF can stimulate AKT activity in breast cancer cells, thereby leading to disassembly of the FAF1/VCP/ β TRCP complex, T β RII membrane accumulation and metastasis, they should also address whether estrogen and progesterone, relevant mitogens that also affect breast cancer cell properties, also lead to AKT activation and FAF1 inhibition.

Response: Many thanks for suggestions; we would like to perform those experiments in the near future but split results into another separate story.

3. Loss of either FAF1 or disassembly of the FAF1/VCP/ β TRCP complex via AKT phosphorylation can each lead to increased breast cancer metastatic potential. Since loss of FAF1 or AKT activation would both lead to increased T β RII levels and TGF- β signaling/metastasis, these events should be redundant and therefore potentially mutually exclusive. The authors should evaluate this possibility in their tissue microarray breast cancer specimens.

Response: In response to reviewer, we evaluated the status of AKT activation (p-AKT 473 staining) and FAF1 expression (FAF1 staining) in the breast cancer tissue microarray specimens. As shown below for reviewer only, FAF1 expression was not significantly reduced in p-AKT 473 high patients, suggesting that the loss of FAF1 and the AKT activation could indeed be independent of each other. Thus the FAF1 phosphorylation by AKT could be critical for breast cancer metastatic potential. We add this as discussion in Page 21. Many thanks for suggestions.

(Low: 0<H score<100; Middle: 100<H score<200; High: 200<H score<300)

4. On page 12 and elsewhere, the authors use a variety of designations for the Faf1 knockout mice, i.e., sometimes in all caps and sometimes with all lower case letters, which should be corrected

Response: They were corrected. Many thanks!

Reviewer #3 (Remarks to the Author):

The manuscript by Zhou et al. entitled 'FAF1 phosphorylation by AKT results in TGF-beta type II receptor accumulation on the cell surface and drives breast cancer metastasis' describes the identification of a novel adapter protein of the TGF-beta type II receptor and the VCP/E3 ligase complex. As such FAF1 turns out to be an important regulator of cell surface TGF-beta type II receptor controlling the activation level of TGF-beta induced signaling. PI3K/AKT signaling antagonizes TGF-beta-induced growth arrest and high levels of TGF-beta in tumours correlate with strong PI3K/AKT signaling. The authors show elegantly that growth factor-induced AKT activity specifically phosphorylates FAF1 resulting in the dissociation of FAF1 from the plasma membrane and its association with the TGF-beta II receptor reinforcing the pro-metastatic functions induced by TGF-beta in different breast cancer models. The authors show that FAF-1 expression levels associate with metastasis free survival of patients. All together the authors position the FAF1-VCP/E3 complex as a controlling point contributing to the shift between normal TGF-beta cytostatic activity towards oncogenic invasive growth and metastasis. The manuscript is well written and experiments are expertly obtained. The following suggestions/remarks are given to expedite publication in Nature communications:

- In Fig 5D the authors explain that in MEFs with graded expression of FAF1, the FAF1^{+/+} MEFs are not expressing EMT inducing transcription factors and are strongly E-cadherin positive. Most MEFs are co-expressing EMT controlling nuclear factors, mesenchymal markers and low E-cadherin expression. The expression bar should be adapted or expression data should be properly normalized and represented as now MEFs are presented as epithelial cells.

Response: The expression data has been normalized to a better internal control and the bar has been re-adapted, many thanks!

- In the discussion the authors mention that Snail is directly regulating the Faf1 promoter which is an important result as it could explain the well-established correlation of EMT inducing transcription factors that facilitate enhanced TGF-beta signaling. These data should be implemented in the manuscript or otherwise the Snail part left out of the discussion. There have been many target genes and differential expressed genes associated with Snail expression published but Faf1 has so far not been reported/listed as a potential candidate target gene. In this context the authors link Snail expression and the general repression of Faf1 in human cancer. This should be tuned down as Snail expression is in most cancer cases hardly expressed and cannot be the reason for the so called broad repression of Faf1.

Response: Many thanks for suggestion; we have left out the Snail part from the discussion.

- Percentage of distant metastasis free survival (Fig 1J) should be explained better in light of the link of FAF1 downregulation and loss of epithelial differentiation which was clearly established by the authors in this study using different breast cancer models. Are the FAF1 low tumors ER-, PR- negative? Furthermore, for human breast cancer besides the potential broad existence of partial EMT, EMT is only clearly detectable in low frequent breast carcinosarcomas and in the molecular subtype of claudin low breast cancers. As the authors link FAF1 with invasion and metastasis through modulation of EMT they should examine if the FAF1 expression level is more

attenuated in these well-established EMT like breast cancers.

Response: Thank you for suggestion, we further explained Fig 1j in the discussion, the end of Page 20. As shown below for reviewer only, we found the FAF1 expression is quite low in triple negative MDA-MB-231 tumors (left panel), which represents subtype of claudin low breast cancers. As shown in the right panel, the western blot comparison between MCF7 and MDA-MB-231 cells showed that FAF1 expression correlates with epithelial marker E-cadherin and inversely correlates with mesenchymal marker Vimentin. We described these results as data not shown in Page 19, in the beginning of the second paragraph.

REVIEWERS' COMMENTS:

Reviewer #1 (Remarks to the Author):

In the revised paper, the authors have addressed all comments by this reviewer. I now find the paper very clear and all minor mistakes seem to have been corrected. I believe that this paper will have a strong impact on the TGF-beta field but also in the broader area of cancer research.

Reviewer #2 (Remarks to the Author):

The authors have satisfactorily addressed my earlier concerns.

Reviewer #3 (Remarks to the Author):

Dear,

- the authors adapted the heatmap representing qRT-PCR of different markers. The expression of E-cadherin looks more acceptable for MEF cells. The answer of the authors is that the expression is normalized to a better internal control - without saying what this control is. Furthermore the material and methods section regarding normalization of the qRT-PCR is completely the same as with the first submission and mentions that the normalization is done only with GAPDH which is a lousy way of normalization. I insist that the authors explain this better what they did to produce the novel version of 5d as it now looks that they rapidly adapted to please the reviewer.

-regarding the general repression of faf1 by the snail transcription factor - the authors followed my comments and left out the idea that snail is driving the general observed faf1 repression in advanced cancers

-Better explanation of Fig1j. A few sentences were added in the discussion - this helps but the added value is limited. Honestly I do not understand why the expression faf1 cannot be supported in a better way for EMT relevant human tumors as the triple negative breast cancers in particular the claudin low cancers.

Showing two cell lines MCF7 and MDAMB231 does not help in this.

Reviewer #3

The authors adapted the heatmap representing qRT-PCR of different markers. The expression of E-cadherin looks more acceptable for MEF cells. The answer of the authors is that the expression is normalized to a better internal control - without saying what this control is. Furthermore the material and methods section regarding normalization of the qRT-PCR is completely the same as with the first submission and mentions that the normalization is done only with GAPDH which is a lousy way of normalization. I insist that the authors explain this better what they did to produce the novel version of 5d as it now looks that they rapidly adapted to please the reviewer.

Response: We apologized for not making proper changes of the qRT-PCR method. We indeed found that the GAPDH is a lousy control for MEF cells and our new analysis is normalized to internal control of 18S RNA levels, the primer sequence are: 18S Forward 5'-gtaaccggtgaaccatt-3'; 18S Reverse 5'-ccatccaatcggtagtagcg-3'. We made proper changes in our method. Thank you for pointing this out for us. We deeply appreciate your help.

-regarding the general repression of faf1 by the snail transcription factor - the authors followed my comments and left out the idea that snail is driving the general observed faf1 repression in advanced cancers

Response: Many thanks for your suggestion. By removing that part, our story gets/reads more focus.

-Better explanation of Fig1j. A few sentences were added in the discussion - this helps but the added value is limited. Honestly I do not understand why the expression faf1 cannot be supported in a better way for EMT relevant human tumors as the triple negative breast cancers in particular the claudin low cancers. Showing two cell lines MCF7 and MDAMB231 does not help in this.

Response: We agree to this reviewer regarding that the EMT is only clearly detectable in low frequent breast carcinomas such as the molecular subtype of claudin low breast cancers.

Figure 1j comes from TCGA breast cancer database. In response to reviewer, we examined in this database whether Claudin-low cancer shows significant difference in FAF1 expression. As shown in volcano plot below, FAF1 expression (mRNA level) is significantly down-regulated in almost any cut-off of Claudin (CLDN1) low cancers. We showed below three different cut-off of CLDN1 mRNA expression by Z score. Many thanks to reviewer for this finding; we showed one of the cut-off as Supplementary Figure S2b. Discussion was accordingly modified.

Z score of CLDN1 < 2.5

Z score of CLDN1 < 1.5

Z score of CLDN1<0.5

Similarly, FAF1 low patients also show significant low CLDN1 level. Volcano plot from the same database was shown below:

However, we did not find positive correlation between FAF1 and ER or PR. As breast cancer is a complex disease, this suggests that the FAF1-low/Claudin-low breast cancers might be the real subtype that is responsive to TGF- β and EMT.